# citrOgen: a synthesis-free polysaccharide and protein antigen-presentation to antibody-induction platform

Joshua L. C. Wong [1,2] ✉, Julia Sanchez-Garrido [1,2], Jaie Rattle[1], Jonathan Bradshaw[1], Vishwas Mishra [1] & Gad Frankel [1] ✉

Existing technologies employed to generate antibodies against bacterial polysaccharides and proteins rely on the availability of purified or synthetic antigens. Here, we present a genetics-based platform that utilises *Citrobacter rodentium* (CR), an enteric mouse pathogen, to both produce and present complex heterologous polysaccharides and protein antigen complexes during natural infection. As proof of concept, we use lipopolysaccharides (O), capsular polysaccharides (K) and type 3 fimbrial (T3F) antigens expressed by the WHO critical priority pathogens *Klebsiella pneumoniae* (KP) and *Escherichia coli* (EC). Following one infection cycle (28 days), CR induces specific IgG antibodies against KPO1, ECO25b, KPK2 and KPT3F. We demonstrate that the antibodies are functional in downstream applications, including protection against pathogenic KP challenge, KP capsular serotyping and KP biofilm inhibition. Whilst KP and EC antigens were used as prototypical examples, this modular platform is now readily adaptable to generate antibodies against diverse polysaccharide and protein antigens, with basic science, public health and therapeutic applications.

The specificity of antibody/antigen (Ab/Ag) binding has been widely leveraged in science, biotechnology, and medicine. Both polyclonal (pAb) and monoclonal (mAb) antibodies are routinely employed across these disciplines and have shaped the face of the modern practice. There is, therefore, significant value in expanding the technologies available to generate new Abs that bind specific Ags of interest.

Abs are employed widely in the field of infectious disease with both diagnostic and, more recently, therapeutic applications. One key use in diagnostics is bacterial serotyping, which has played important roles in pathogen epidemiology. Therapeutic Abs are licensed for both the prevention and treatment of viral and bacterial infections in humans. Palivizumab, a humanised monoclonal IgG Ab that binds Respiratory Syncytial Virus, was the first mAb licensed for the prophylaxis of infectious disease in 1998[1]. Bezlotoxumab, a mAb against the enteric pathogen *C. difficile*, the sole mAb licensed in bacterial

disease, was licensed in 2016 for the secondary prevention of *C. difficile* colitis[2]. More recently, during the SARS-CoV-2 pandemic, both pAb administration (convalescent sera)[3] and mAbs against spike protein (e.g., bamlanivimab, etesevimab, casirivimab, and imdevimab)[4] were used in the prevention and treatment of COVID-19 disease.

We currently have limited therapeutic options to combat the antimicrobial resistance (AMR) pandemic with both anti-bacterial Abs (α-bAbs) and vaccines representing promising modalities to tackle this contemporary threat. Vaccines rely on inducing an enduring memory response through the immunisation of naïve populations with target bacterial Ags. This allows adaptive host immunity to briskly reactivate upon subsequent pathogen exposure, clearing infection and reducing disease severity. In the case of extracellular bacteria, the effector arm of this response is centrally mediated by Abs. In contrast, α-bAb therapeutics are produced in vitro and administered to patients as prophylaxis or in treatment settings. Uniquely, Fc-enhanced effector

[1]Department of Life Sciences, Imperial College London, London, UK. [2]These authors contributed equally: Joshua L. C. Wong, Julia Sanchez-Garrido.
✉e-mail: joshua.wong@imperial.ac.uk; g.frankel@imperial.ac.uk

functions can be molecularly introduced to α-bAbs that prolong intravascular half-life or promote hexamerisation and complement-mediated cytotoxicity[5,6]. There are merits and disadvantages to both approaches, for example, whilst α-bAb administration ensures that the effector arm of the host response is delivered directly to the patient, it fails to induce immunological memory. However, memory responses are often blunted in the high-risk populations that would benefit most from vaccination, such as the elderly[7]. As such, both approaches are being actively pursued and could be used in combination or to target different populations.

In target-specific α-bAb campaigns, Abs that bind predetermined Ags are generated and selected for downstream applications. Target choice can be informed a priori by pathogen epidemiology and genomics, selecting bacterial Ags that are both widespread and conserved across a bacterial species. Ags representing virulence determinants can also be selectively targeted as Ab-binding has the potential to functionally disrupt pathogenic mechanisms[8]. The immunisation of experimental animals or human subjects is a common method to generate target-specific Abs. This results in sera containing pAb secreted from reactive B-cells. These cells can be immortalised or Ab-encoding regions subcloned to produce mAbs in vitro. Irrespective of host species or Ag, the generation of such responses is fundamentally reliant on a sufficient source of conformationally correct immunogen. Generating these immunogens represents a key bottleneck in both target-specific Ab generation[9] and vaccine development.

Polysaccharide Ags are diverse and typically composed of variable sugar subunits, each covalently connected in the correct order by specific glycosidic links. These linkages can be linear or include branching arrangements that determine the final polysaccharide structure. In isolation, polysaccharides typically require carrier protein glycoconjugation to generate formulations that are sufficiently immunogenic[9]. Production of sufficient quantities of protein Ags, which are typically affinity purified following de novo synthesis in cells, is largely restricted to protein monomers and not applicable to multiprotein complexes. The correct formation of the latter not only requires the presence of chaperones and signal sequences, directing subunit assembly in the correct subcellular space, but also necessitates conformationally correct monomers to interact with each other in the correct stochiometric ratio. Bacterial genomes have compartmentalised the genes required to synthesise both polysaccharide and multimeric protein Ags into operons, providing cloneable linear DNA fragments that we exploit in the development of the technology we describe here.

*Citrobacter rodentium* (CR) causes colitis in mice with an infection strategy mirroring the human pathogen Enteropathogenic *Escherichia coli* (EPEC). CR attaches intimately to intestinal epithelial cells (IECs), forming attaching and effacing lesions and actin-rich pedestal-like structures, a prerequisite for productive infection and colitis[10]. A major virulence factor enabling CR infection is the type 3 secretion system (T3SS)[11], which translocates 31 effector proteins into IECs, facilitating CR adherence and proliferation[12]. The resolution of CR infection coincides with the onset of IgG secretion[13,14]. However, until this adaptive immune response clears the infection, CR maintains high mucosal-associated burdens, inducing significant local inflammation with associated immune cell recruitment[12]. We hypothesised that this natural host-pathogen context represents an evolutionarily pre-optimised environment to present alternative (heterologous) Ags to the mouse host, thereby generating mature, target-specific Ab responses. This hypothesis forms the basis of our platform, which we term citrOgen.

Heterologous Ag delivery employing alternative bacterial chassis has been evaluated in previous studies and largely focussed on live attenuated vaccine candidates. Common candidates include *Salmonella* Typhimurium and *Shigella spp*, where heterologous Ag expression aims to broaden the vaccines range to now include pathogens beyond the attenuated candidate itself[15–17]. This field has been hampered by the low stability of heterologous Ags in vivo and failure to induce protective responses. Indeed, when the licensed live-attenuated *Salmonella* Typhi vaccine strain Ty21a expressing *Shigella sonnei* O2a Ag was administered to human subjects, only 19–27% ($n = 121$ participants) had detectable *S. sonnei* IgG responses[18]. A subgroup of participants and controls then underwent *S. sonnei* challenge and clinical shigellosis developed at the same rate in both groups, providing no evidence of protection or amelioration of disease.

Here we present citrOgen, a flexible and modular in vivo Ag presentation platform co-opting CR to present heterologous Ags. CitrOgen represents a single unifying genetics-based system that can be used to generate functional target-specific α-bAb responses to divergent major Ag subtypes in mice. We select four structurally diverse and validated Ags from the WHO critical priority pathogens *Klebsiella pneumoniae* (KP) and *E. coli* (EC) to test the platform[19]. The choice includes two surface polysaccharide families, lipopolysaccharide (LPS–KP O1 & EC O25b) and capsular polysaccharide (CPS–KP K2) and a protein complex, type 3 fimbriae (KP T3F). Substituting the endogenous CR O152 LPS O-Ag with KP O1 results in lower gut colonisation, which we overcome by overexpressing the CR T3SS effector EspO. Functionally, EspO-augmented α-KP O1 Ab responses potently protect mice from sepsis-induced organ failure following KP challenge. Furthermore, we are able to rapidly modify CR to express EC O25b, eliciting specific α-EC O25b IgG responses after a single infection cycle (28 days) in mice. We then successfully generate α-CPS Abs, which we employ to serotype a diverse KP collection of 100 clinical isolates. Finally, we demonstrate citrOgen's ability to raise Abs to a large cell surface filamentous structure, KP T3F, the regulated expression of which we design to coincide with CR attachment to host epithelia in vivo. These α-KP T3F Abs effectively block T3F-mediated KP biofilm formation. Together, these results demonstrate the ability to rapidly generate functional, target-specific Abs against a broad range of major bacterial Ags.

## Results

### Heterologous KP O1 O-Ag expression causes an in vivo colonisation defect

Our goal was to express diverse heterologous Ags in CR to generate target-specific Abs in mice. The *rfb* locus typically encodes all the genes required for O-Ag synthesis and export in Gram-negative bacteria[20]. An exception is the KP O1 Ag, where *rfb*-derived O2 Ag is modified by the glycosyltransferase activity of two enzymes, WbbY and WbbZ, which are encoded at a distinct chromosomal position[21]. Moreover, KP employs an ABC-transporter instead of the more common *wzy*-dependent O-Ag export process used by $CR_{WT}$[22].

We started by identifying the *rfb* locus in $CR_{WT}$, which is encoded on the reverse strand and flanked by *galF* (5′) and *gnd* (3′) genes (Fig. 1A). We used in silico serotyping to confirm that $CR_{WT}$ encodes an *E. coli*-like O152 Ag[23]. The endogenous O152 genes were replaced with those comprising the KP O2a *rfb* locus (cloned en bloc) by homologous recombination and the *wbbYZ* gene cluster inserted into the CR genome by Tn7 transposon mutagenesis (Fig. 1A)[24]. Tn7 inserts at a neutral insertion site, downstream of the *glmS* site (*attTn7* site) and we have already used this approach to successfully express multiple different cargos in vivo[25]. This strain, $CR_{KPO1}$, contains all the genetic determinants required to synthesise and export KP O1 Ag, albeit using a non-native ABC-transporter export pathway. We did not modify the CR gene products that synthesise lipid A or core oligosaccharide (Fig. 1B) and, therefore, $CR_{KPO1}$ is predicted to produce a hybrid LPS molecule; the endogenous CR Raetz pathway and CR *rfa* locus producing lipid A and core regions, respectively, with KP O1 Ag attached to the native CR lipooligosaccharide scaffold (Fig. 1B). We purified LPS from $KP_{WT}$,

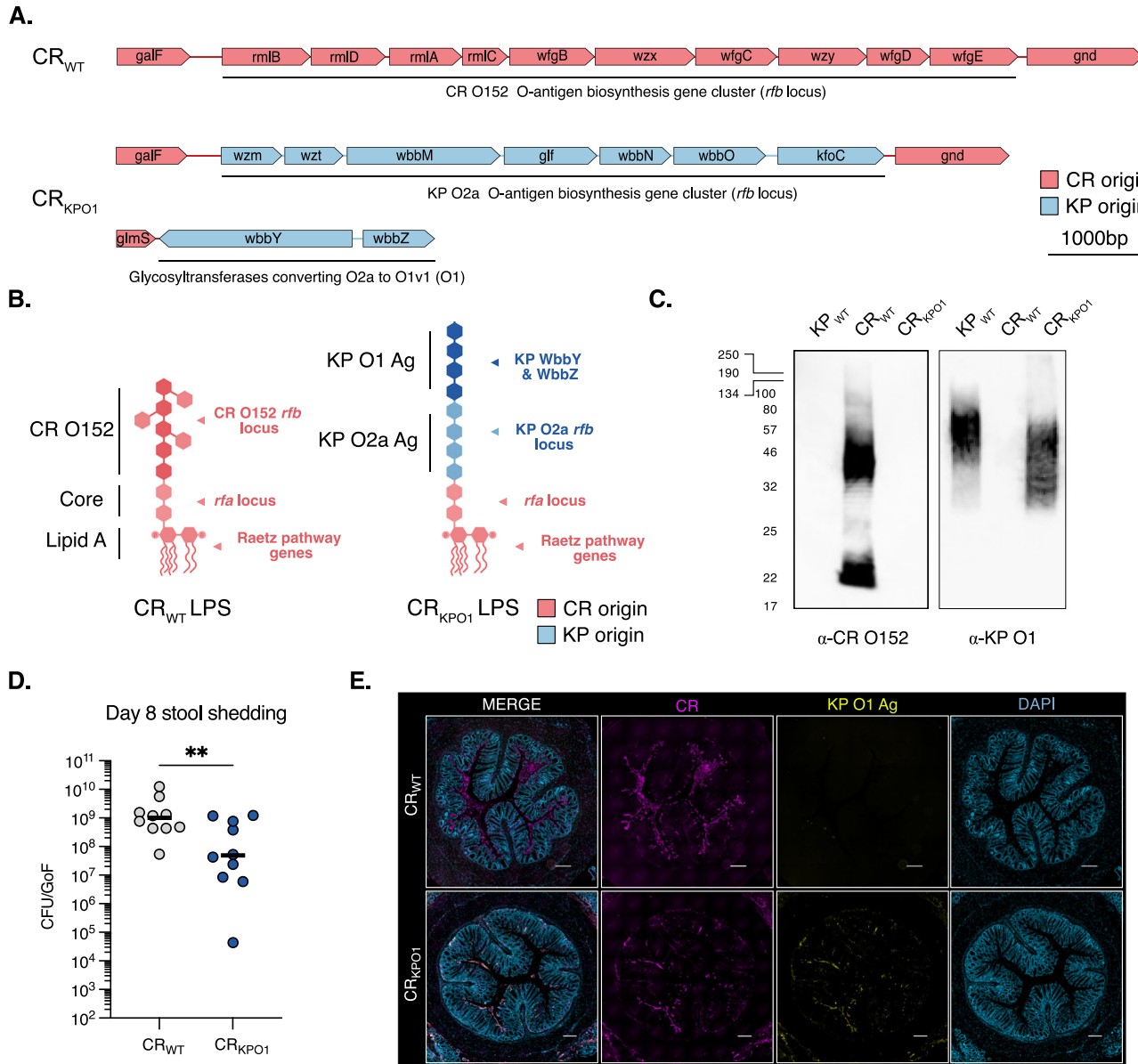

**Fig. 1 | Heterologous KP O1 O-Ag expression causes and in vivo colonisation defect. A** The *rfb locus* in CR_{WT} encodes a O152-like *E. coli* O-Ag locus located between *galF* and *gnd*. In CR_{KPO1} the CR_{WT} *rfb* locus was substituted with the genes encoding KP O2a O-Ag and *wbbYZ* (converted KP O2a to KP O1v1) inserted at the 3′ *glmS* site. **B** CR_{KPO1} expresses a hybrid lipopolysaccharide molecule with native CR lipid A and core lipooligosaccharide linked to KP O1 Ag. Created using BioRender. Frankel, G. (2025) https://BioRender.com/8d48gy1. **C** Western blot confirmation of heterologous KP O1 Ag in CR_{KPO1} performed on purified LPS preparations. α-O152 Ab are immunoreactive with CR_{WT} but not KP_{WT} or CR_{KPO1} whereas α-KP O1 mAb (C13) is immunoreactive with both KP_{WT} and CR_{KPO1}. Molecular weights (in kDa) are shown on the left. Representative of $n = 2$ biological repeats. **D** At 8 dpi following oral gavage, faecal CFUs are significantly lower in CR_{KPO1} compared with CR_{WT}. $n = 10$ mice per group, 2 biological repeats. Central tendency: median. Two-sided Mann–Whitney test. **, $p < 0.01$. GoF, gram of faeces. **E** Colonic histology was taken 8 dpi and stained for CR, KP O1 Ag and DAPI staining used as a structural marker. CR_{WT} is observed confluently adhering to the luminal surface of the colon with no KP O1 Ag expression detectable. Sparse staining for CR_{KPO1} is observed however, where present, KP O1 Ag expression detectable. Scale bar = 200 μm. Images are representative of 3 biological repeats. Source data and exact *p*-values are provided in the Source Data file.

CR_{WT} and CR_{KPO1} and confirmed the presence of a substituted O-Ag CR_{KPO1} by silver stain, with a banding pattern distinct from CR_{WT} and similar to KP_{WT} (Supplementary Fig. 1A). The specificity of O-Ag substitution was confirmed Western blot. As anticipated α-O152 pAb sera recognised CR_{WT} but was not reactive against KP_{WT} and CR_{KPO1} (Fig. 1C). We then used an α-KP O1 mAb (C13)[26], which bound to both KP_{WT} and CR_{KPO1}, confirming heterologous O-Ag expression in CR. The homologous recombination vector employed to substitute the KP O2 locus was designed to include the endogenous 5′ CR promoter, ensuring that future O-antigen locus substitutions can be achieved by using the same vector, simply modified with O-Ag genes of choice.

We tested if replacing CR O152 with KP O1 affected infectivity in vitro and confirmed that CR_{KPO1}, like CR_{WT}, induced actin-rich pedestal formation following the infection of mouse fibroblasts cells. Furthermore, detectable KP O1 Ag expression was restricted to CR_{KPO1} (Supplementary Fig. 1B). We therefore proceeded to infect CD-1 mice with $10^9$ colony-forming units (CFUs) of CR_{WT} or CR_{KPO1} by oral gavage (Fig. 1D). At 8 days post-infection (dpi), when stool CR shedding peaks, faecal CR_{KPO1} CFU counts were significantly lower than CR_{WT} (Fig. 1D). Staining of colonic sections revealed that whilst CR_{KPO1} maintained heterologous KP O1 expression in vivo, there were substantially fewer adhered bacteria compared to CR_{WT} (Fig. 1E and Supplementary

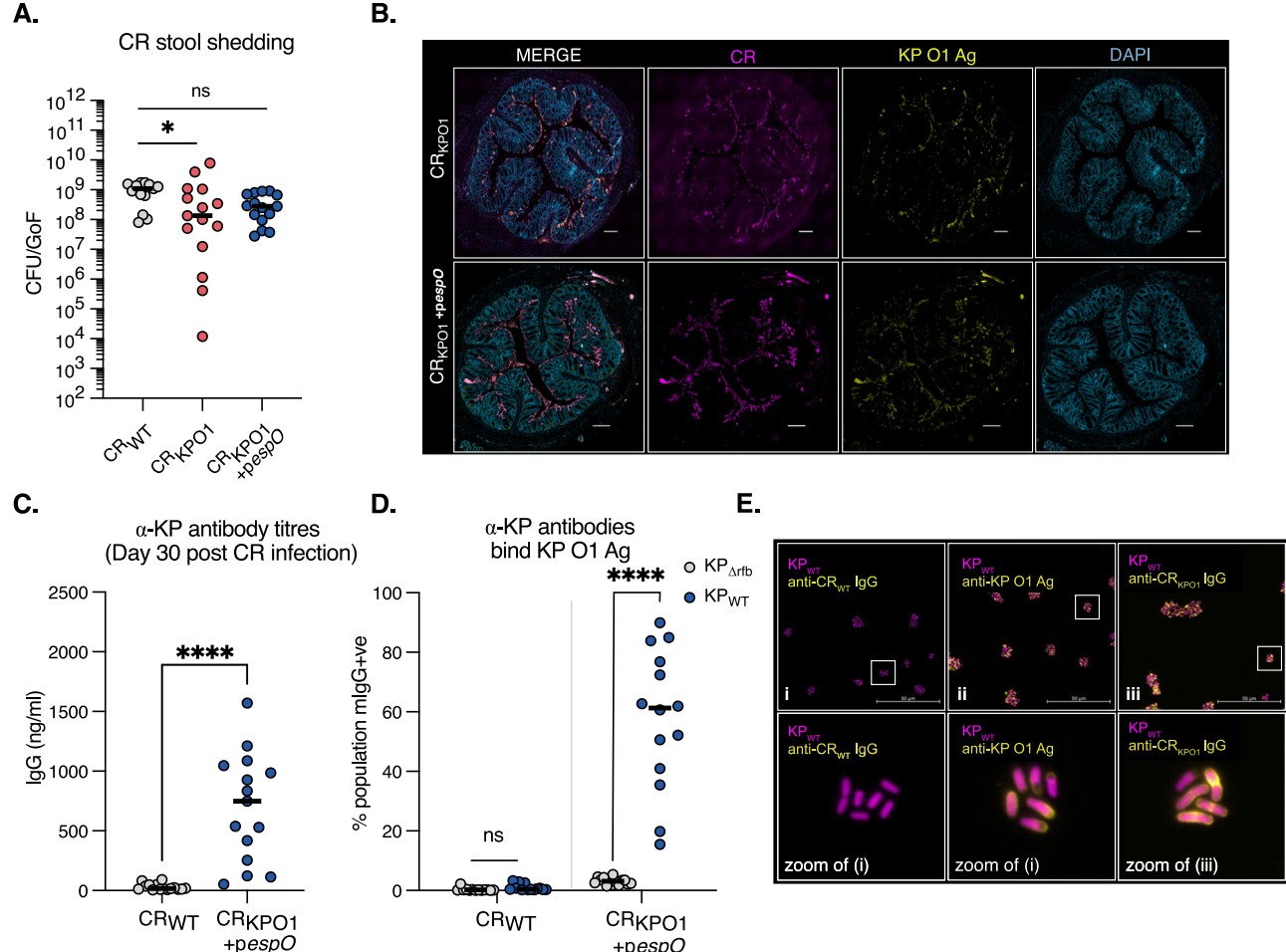

**Fig. 2 | CR_KPO1 overexpressing EspO maintains infectivity and induces specific anti-KP O1 Ag IgG responses. A** At 8 dpi *espO* overexpression in CR_KPO1+p*espO* reverses the colonisation defect observed in CR_KPO1 restoring colonisation to that of the parental CR_WT strain. Central tendency: median. Ordinary one-way ANOVA with Dunnett's multiple comparison. **B** Colonic histology was taken 8 dpi and stained for CR, KP O1 Ag and DAPI staining used as a structural marker. CR_KPO1+p*espO* now confluently adhering to the luminal surface with KP O1 Ag expression detectable and co-localising to CR staining. Sparse staining is observed for CR_KPO1 with KP O1 Ag expression. Scale bar = 200 μm. Images are representative of 3 biological repeats. **C** At 30 dpi blood was taken from the tail vein of mice that cleared CR_WT and CR_KPO1+p*espO*. ELISA, using heat-killed KP_WT (expressing KP O1 Ag), revealed α-KP_WT IgG in mice following infection with CR_KPO1 +p*espO* but not CR_WT controls. Central tendency: mean. Two-sided Mann–Whitney test. **D** Sera

from mice was tested for binding to live KP_WT and KP_Δrfb. Binding was assessed by flow cytometry and expressed as the percentage of the population positive for mouse IgG (mIgG+ve). Sera from CR_WT-infected mice demonstrated absent binding to either KP_WT or KP_Δrfb, whereas sera from CR_KPO1+p*espO* -infected selectively bound KP_WT expressing KP O1 Ag. Central tendency: mean. Two-sided, unpaired *T*-test. **E** Sera from CR_WT and CR_KPO1+p*espO* -infected mice was used to stain KP_WT and image by immunofluorescence microscopy. An α-KP O1 Ag mAb (C13) was used as a positive control for KP O1 Ag staining. CR_WT derived sera results in no detectable staining of KP_WT whereas sera from CR_KPO1 +p*espO* -infected mice and the control C13 mAb phenocopy each other with membrane-associated staining of KP_WT. **A**, **C**, **D**. Data from 3 biological repeats; *n* = 15 mice per group (**A** and **C**) or *n* = 13 (CR_WT) or 14 (CR_KPO1+p*espO*) (**D**). ns non-significant; *, *p* < 0.5; ****, *p* < 0.0001. Source data and exact *p*-values are provided in the Source Data file.

---

Fig. 1C). These data indicated an in vivo colonisation defect imposed by heterologous Ag expression that needed to be addressed to develop an optimal platform.

## EspO overexpression restores CR_KPO1 colonisation

A mutant in the *Shigella flexneri* T3SS effector *ospE*, a homologue of the CR effector *espO*, results in a colonisation defect in a guinea pig gut infection model, as OspE suppresses epithelial detachment[27]. We therefore hypothesised that over expression of EspO would promote greater bacterial attachment to IECs during infection. Indeed, while deletion of *espO* in CR_WT showed a tendency towards decreased proportion of CR-associated IECs, overexpression of EspO increased the proportion from 14.6% to 22.9% (Supplementary Fig. 2A–C). We then overexpressed EspO in CR_KPO1 (CR_KPO1+p*espO*) and infected CD-1 mice; infections with CR_WT and CR_KPO1 were used as controls. Enumeration of faecal CFUs now revealed no difference between the faecal CFUs between CR_WT and CR_KPO1+p*espO* at 8 dpi (Fig. 2A). Staining of colonic

sections demonstrated higher levels of CR_KPO1+p*espO* adhesion to the epithelia compared to CR_KPO1, which maintained heterologous KP O1 expression in vivo (Fig. 2B).

## CR_KPO1+p*espO* infection elicits KP O1 specific IgG antibodies

We inoculated mice with CR_KPO1+p*espO* and CR_WT by oral gavage. After CR clearance (c. 28 dpi), we collected blood and assessed α-KP_WT IgG antibody titres by enzyme-linked immunosorbent assay (ELISA), using a capture plate coated with heat-killed KP_WT. While control CR_WT mice sera had negligible α-KP_WT IgG binding, this increased significantly in sera from CR_KPO1+p*espO*-infected mice, indicating that we had successfully raised α-KP_WT Abs (Fig. 2C). To confirm the specificity of the Ab response, we used an isogenic rough (absent O-Ag) KP mutant, KP_Δrfb, and assessed binding to live bacteria by flow cytometry (Fig. 2D). Consistent with the ELISA data, sera from CR_WT-infected mice bound to neither KP_WT nor KP_Δrfb. However, sera from CR_KPO1+p*espO*-infected mice bound specifically to KP_WT and not KP_Δrfb, confirming

the presence of α-KP O1 Abs (Fig. 2D and Supplementary Fig. 2D). We then assessed the pattern of binding to $KP_{WT}$ by immunofluorescence (IF) microscopy (Fig. 2E). This confirmed the same pattern of binding of $CR_{KPO1}$+p*espO* sera and the reference α-KP O1 Ab C13. No detectable signal was observed in sera from $CR_{WT}$-infected mice.

## Anti-KPO1 IgG Abs are functionally protective against KP pulmonary challenge

We next tested the functionality of the $CR_{KPO1}$+p*espO*-induced α-KP O1 pAbs by lung challenge with $KP_{WT}$. Mice that had been infected and cleared $CR_{KPO1}$+p*espO* were intubated and administered 500 CFU of $KP_{WT}$ directly into the lungs. In preliminary experiments with CD-1 mice, this $KP_{WT}$ inoculum was found to result in a progressive increase in KP CFUs in the lungs and blood, resulting in a severe disease phenotype at 72 h post-infection (hpi) (Supplementary Fig. 3A–C). Mice that had been infected by and subsequently cleared $CR_{WT}$ were also challenged as controls. We evaluated infection outcomes by assessing $KP_{WT}$ replication and protection against host organ failure employing markers of haematological, lung and renal injury in line with international consensus[28,29]. At 72 hpi $KP_{WT}$ burdens in the lungs of $CR_{WT}$-preinfected mice reached a median of c. $10^9$ CFUs (Fig. 3A). In contrast, $CR_{KPO1}$+p*espO*-preinfected mice had significantly reduced lung CFUs with $KP_{WT}$ only detectable in the lungs of 30% (5/14) of mice. KP bacteraemia was absent in all $CR_{KPO1}$+p*espO*-preinfected mice in comparison to 69% (9/13) of $CR_{WT}$-preinfected control mice (Fig. 3B).

We then assessed if the reduction of $KP_{WT}$ burden translated into a positive effect on host physiology. Only 3 mice (20%, 3/15) in the $CR_{KPO1}$+p*espO*-preinfected group lost more than 10% of their starting weight when challenged with $KP_{WT}$, whereas all mice in the $CR_{WT}$-preinfected group lost weight after KP challenge, with 86% (12/14) losing more than 10% of their starting weight (Fig. 3C). Total white cell counts, lymphocyte counts and platelets in the blood demonstrated significantly positive improvements in $CR_{KPO1}$+p*espO*-preinfected mice compared to $CR_{WT}$-preinfected animals (Fig. 3D–F). Specifically, the extent of sepsis-induced leucopaenia, lymphopaenia and thrombocytopaenia were abrogated by the $CR_{KPO1}$+p*espO* pre-infection, while the number of neutrophils did not change between the mouse groups (Fig. 3G). Moreover, the positive acute phase reactants (APR) Serum amyloid A and C-reactive protein were significantly lower in $CR_{KPO1}$+p*espO*-preinfected mice compared $CR_{WT}$-preinfected animals (Fig. 3H, I). This beneficial modulation in APRs translated into higher levels of the negatively regulated APR, albumin, in $CR_{KPO1}$+p*espO* preinfected mice (Fig. 3J).

We then used three markers to assess lung injury: IL1β and myeloperoxidase (MPO) in lung homogenate and surfactant protein D (SPD) levels in the serum[28]. This revealed that $CR_{KPO1}$+p*espO* preinfection resulted in significant reductions in IL1β and SPD levels compared to $CR_{WT}$ preinfection, indicating reduced lung inflammation and improved maintenance of the alveolar capillary barrier respectively (Fig. 3K, L). Preinfection with $CR_{KPO1}$+p*espO* also resulted in significantly reduced neutrophilic alveolitis, evidenced by reduced lung MPO levels compared to $CR_{WT}$-preinfected mice (Fig. 3M). Finally, we assessed acute kidney injury by measuring serum creatinine and cystatin C levels. While high levels of creatinine and cystatin C were found in $KP_{WT}$ challenged $CR_{WT}$-preinfected mice, improvements in both excretory renal function markers were seen in $CR_{KPO1}$+p*espO*-preinfected mice (Fig. 3N, O). These data indicate that α-KP O1 Abs raised by citrOgen were protective against KP challenge, restricting both bacterial replication and dissemination, and preventing KP sepsis-induced organ failure.

## citroGen can be rapidly adapted to present alternative O-Ags

Having overcome the fitness cost imposed by heterologous O-Ag expression in CR and demonstrated functional responses, we wanted to further test the platform's modular nature and efficiency. To that end, we selected *E. coli* O25b Ag (EC O25b), frequently encoded by ESBL-producing ST131 *E. coli* strains (EC958 representing the prototype[30]), and used our vector system to generate $CR_{ECO25b}$ (Fig. 4A). O25b Ag was selected for two reasons, firstly, it is expressed by an alternative WHO critical priority pathogen and, secondly, it is exported by the most common wzy-dependent export pathway. The additional gene, *wzz*, which lies outside the *rfb* locus, is a chain length determinant in wzy-dependent systems; we did not replace this and opted to keep the native $CR_{WT}$-encoded variant. We confirmed specific O25b expression by agglutination, using reference α-O25b pAb sera, which agglutinated EC958 and $CR_{ECO25b}$ but not $CR_{WT.}$ (Fig. 4B). Silver staining of LPS preparations confirmed the presence of O-Ag in $CR_{ECO25b}$, which now runs at a higher molecular weight, as dictated by the novel wzz(CR)/wzy(EC958) interaction (Supplementary Fig. 4). Mice were orally infected with $CR_{ECO25b}$+p*espO* and at 8 dpi no colonisation defect was observed compared to $CR_{WT}$ (Fig. 4C). Blood was collected from mice after CR clearance and sera was used in an ELISA against EC958 and an isogenic rough mutant (EC958$_{\Delta rfb}$). Absent Ab responses were detected from $CR_{WT}$-infected mice against either EC958 or EC958$_{\Delta rfb}$. However, sera from mice infected with $CR_{ECO25b}$+p*espO* specifically reacted to EC958, with absent binding to EC958$_{\Delta rfb}$ (Fig. 4D). This specificity confirmed that citrOgen can be used against additional O Ags, irrespective of export pathway. Moreover, it confirms that Ag selection to antibody production is achievable in c. 35 days.

## citroGen presentation of heterologous KP capsular polysaccharide Ag

We next explored expanding the citrOgen platform to generate α-CPS Abs. $KP_{WT}$ produces K2 CPS, encoded by a c. 22 Kb K2 *cps* locus, lying immediately upstream of the *rfb* locus in the $KP_{WT}$ genome. As upstream of the CR O152 *rfb* locus lie the genes involved in synthesis and export of the extracellular polysaccharide colanic acid (Fig. 5A), we replaced the $CR_{WT}$ genes between *ROD21991* and *galF* (c. 24 kb) with the KP KL2 operon from *wzi* to *manB* by homologous recombination, generating strain $CR_{KPK2}$+p*espO* (Fig. 5A). We chose this insertion site as we anticipate that it will tolerate other *cps* locus insertions in future applications of this technology. This established vector can be used as the template together with *cps* locus of choice (e.g., KP *cps* loci vary from 10 to 30 kb and include internal promoters[31]).

We infected mouse fibroblasts to evaluate if the expression K2 capsule impacted on CR T3SS functionality. $CR_{KPK2}$ formed actin-rich pedestals and heterologous K2 expression was confirmed by staining with reference K2 antiserum (Supplementary Fig. 5A). This suggests that the K2 capsule does not obstruct the T3SS needle, likely due to the fact that it is extended by a long EspA filament[32]. We confirmed that EspO overexpression was required for optimal colonisation in the context of heterologous CPS expression, as mice orally infected with $CR_{KPK2}$ colonised at a lower level compared to $CR_{WT}$ at 8 dpi (Supplementary Fig. 5B). We then proceeded to infect CD-1 mice with $CR_{KPK2}$+p*espO*, or $CR_{WT}$ as a control, which were shed at comparable levels (Fig. 5B). Following $CR_{KPK2}$+p*espO* and $CR_{WT}$ clearance, we collected blood samples and α-KP IgG levels were assessed by ELISA, using $KP_{WT}$ and an acapsular *wcaJ* mutant, $KP_{\Delta wcaJ}$. No binding to either $KP_{WT}$ or $KP_{\Delta wcaJ}$ was observed with sera from $CR_{WT}$-infected mice. In contrast, sera from mice infected with $CR_{KPK2}$+p*espO* bound to $KP_{WT}$ but not to $KP_{\Delta wcaJ}$ (Fig. 5C). This indicated that KP K2 was successfully presented by the citrOgen platform in vivo and that α-KP IgG was specifically raised against the heterologous KP K2 antigen.

KP strains demonstrate broad diversity in CPS Ags with over 147 described to date in circulating strains[31]. However, despite their key role in virulence and frequent use in genome-based epidemiology, no reference antisera are currently being produced for in vitro applications. We used sera derived from $CR_{KPK2}$+p*espO* infection, together with a reference K2 antiserum as a control, to stain $KP_{WT}$. No detectable binding was seen in sera from $CR_{WT}$-infected mice. In contrast, binding was seen when $KP_{WT}$ was stained with the $CR_{KPK2}$+p*espO* sera,

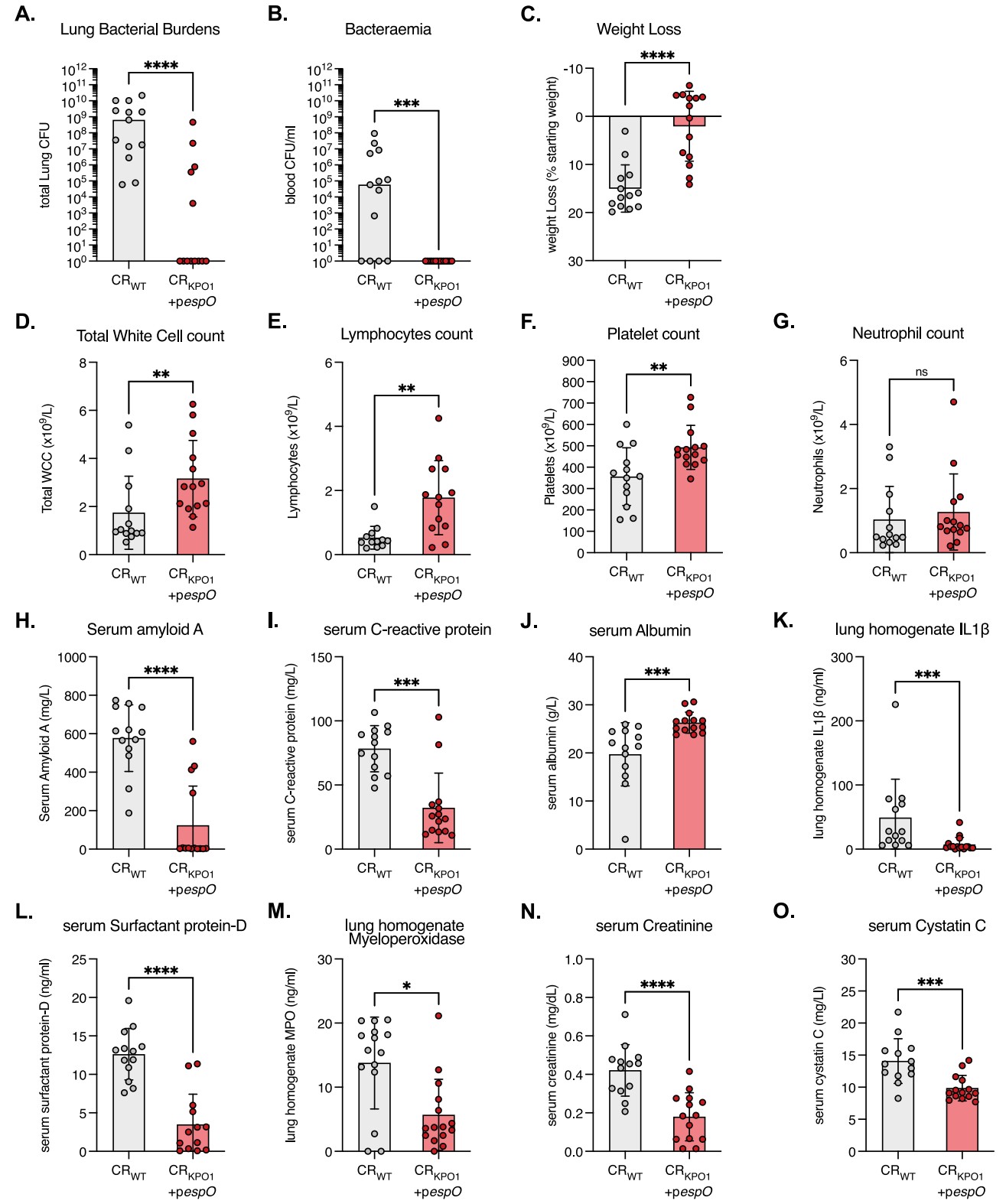

which phenocopied the pattern of binding seen with the reference K2 antiserum (Fig. 5D). Together with the ELISA results, this provides strong evidence for specific heterologous α-KP K2 Abs induced by CR$_{KPK2}$+p$espO$ infection.

### KP serotyping using the citrOgen KP K2 sera
We next tested if the citrOgen KP K2 sera could be used for serotyping. To this end, we used a publicly available collection of 100 KP isolates,

representing 54 distinct CPS types determined by genomic analysis[33]. We incubated the strains with citrOgen KP K2 sera and analysed binding by flow cytometry (Fig. 5E and Supplementary Fig. 5C). We included KP$_{WT}$ and KP$_{\Delta wcaJ}$ as controls and measured the median fluorescence intensity (MFI) across the collection. The citrOgen KP K2 sera specifically bound KL2 isolates and KP$_{WT}$ with minimal cross reactivity and absent binding to KP$_{\Delta wcaJ}$. This binding was comparable to that observed when staining the KL2 strains with our reference K2

**Fig. 3 | Antibodies to KP O1 O-antigen protect against KP pulmonary challenge.** Mice that had cleared pre-infection with $CR_{WT}$ and $CR_{KPO1}+pespO$ were challenged by intratracheal infection with 500 CFU of $KP_{WT}$. At 72 hpi we analysed KP replication (**A, B**) and host-derived severity indicators (**C–O**). $n = 15$ mice per group (starting) over 3 biological repeats. **A, B** $KP_{WT}$ replication and bacteraemia are significantly reduced in the lungs (**A**) and blood (**B**) of $CR_{KPO1}+pespO$-preinfected mice compared to $CR_{WT}$ controls. Central tendency: median. Two-sided Mann–Whitney test. **C** $KP_{WT}$ induces significant weight loss in $CR_{WT}$-preinfected mice, which is reversed in $CR_{KPO1}+pespO$ preinfection. Central tendency: mean, error: standard deviation (SD). Two-sided Mann–Whitney test. **D–F** Clinical haematological markers of disease severity are significantly improved in $CR_{KPO1}+pespO$-preinfected mice compared to $CR_{WT}$ controls. Total white cell counts (**D**), lymphocyte counts (**E**) and platelets counts (**F**) are significantly lower in $CR_{WT}$-preinfected controls, indicating increased diseases severity. The innate responders, neutrophils (**G**) are unchanged between the $CR_{WT}$ and $CR_{KPO1}+pespO$ preinfected groups. Central tendency: mean, error: SD. Two-sided unpaired $T$-test. **H–J**. The acute phase reactants (Serum amyloid A (**H**), C-reactive protein (**I**) and Albumin (**J**)) demonstrate a reduced inflammatory state in $CR_{KPO1}+pespO$-preinfected mice compared to $CR_{WT}$ controls. Central tendency: mean, error: SD. **H** and **J** Two-sided Mann–Whitney test, **I** Unpaired $T$-test. **K–M** Acute lung injury markers (lung homogenate IL1β (**K**), serum surfactant protein D (**L**) and lung homogenate myeloperoxidase (**M**)) are improved by pre-infection with $CR_{KPO1}+pespO$ compared to $CR_{WT}$ preinfection. Significant reductions in lung IL1β, improvement in the alveolar capillary barrier (surfactant protein D) and neutrophilic alveolitis (myeloperoxidase) indicate α-KP O1 Abs protect against lung injury. Central tendency: mean, error: SD. **K** and **M** Two-sided Mann–Whitney test, **L** Two-sided unpaired $t$-test. **N, O** Acute kidney injury and excretory renal failure are reduced in mice preinfected with $CR_{KPO1}+pespO$ compared to $CR_{WT}$. Serum creatinine (**N**) and serum cystatin C (**O**) are reduced in $KP_{WT}$ infected mice who have cleared preinfection with $CR_{KPO1}+pespO$ compared to $CR_{WT}$. Central tendency: mean, error: SD. Two-sided unpaired $t$-test. ns non-significant; *, $p < 0.5$ **; $p < 0.01$, ***, $p < 0.001$; ****, $p < 0.0001$. Source data and exact $p$-values are provided in the Source Data file.

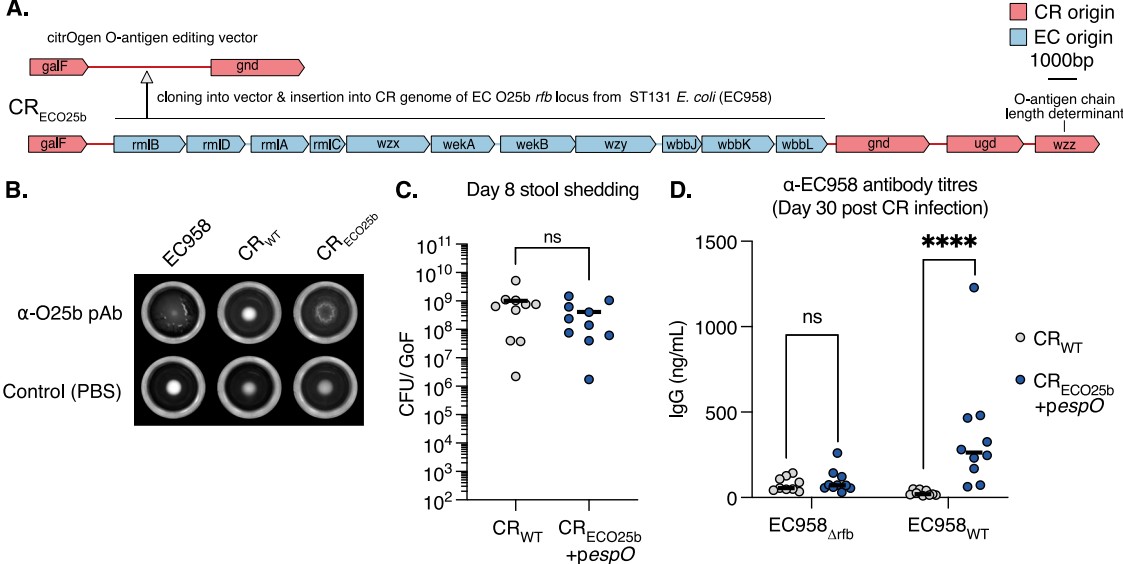

**Fig. 4 | Rapid citrOgen modification to present *E. coli* O25b O-Ag. A** The CR *rfb locus* editing vector can be rapidly modified to insert the O25b *rfb* locus from EC958 (ST131 ESBL-producing strain) into the CR genome, generating $CR_{ECO25b}$. **B** Agglutination of EC958 and $CR_{ECO25b}$ with polyclonal sera to O25b O-antigen. $CR_{WT}$ forms a tight pellet, indicating no agglutination of this strain. In control wells (PBS), all strains form tight pellets in the absence of agglutination. **C** At 8 dpi following oral gavage, faecal CFUs are not significantly lower in $CR_{ECO25b}+pespO$ compared with $CR_{WT}$. $n = 10$ mice per group, 2 biological repeats. Central tendency: median. Two-sided unpaired $T$-test. GoF, gram of faeces. **D** At 30 dpi, blood was taken from mice that cleared infection with $CR_{WT}$ or $CR_{ECO25b}+pespO$. An ELISA was conducted against heat-killed $EC958_{WT}$ (expressing O25b) and an isogenic rough mutant ($EC958_{\Delta rfb}$). $CR_{WT}$ infected mice had no detectable IgG responses to $EC958_{WT}$ or $EC958_{\Delta rfb}$. $CR_{ECO25b}+pespO$ infected mice had IgG responses specifically against $EC958_{WT}$. $n = 10$ mice per group, 3 biological repeats. Central tendency: mean. Two-sided unpaired $t$-tests. ns non-significant; ****, $p < 0.0001$. Source data and exact $p$-values are provided in the Source Data file.

antiserum (Fig. 5E and Supplementary Fig. 5D). To confirm the KL assignments, we selected 9 isolates for IF with the reference K2 antisera (Fig. 5F). We observed a pattern of binding consistent with capsule in the KL2 strain but not the surrounding non-KL2 strains. We concluded that the citrOgen KP K2 sera can be used for serotyping and validated the collection's KL2 assignments.

**citroGen presentation of heterologous protein complex Ags**
We next assessed the utility of citrOgen in generating Abs against heterologous proteins, using the well-conserved KP T3F as a model protein complex[34]. T3F are a KP virulence factor that mediate binding to abiotic surfaces (such as plastic and glass) and are involved in the first steps of biofilm formation. The major T3F subunit is MrkA which together with MrkD (adhesin tip) and MrkF (unknown function) are externalised via a β-barrel transmembrane usher protein, MrkC, and assemble into T3F on the KP cell surface (reviewed in Murphy et al. 2012[35]). In the periplasm, a chaperone, MrkB, binds MrkA to prevent ectopic pilin polymerisation. Therefore, MrkABCDF are all required for T3F biogenesis. T3F expression is tightly regulated in KP by an adjacent operon (*mrkHIJ*) via the second messenger cyclic-di GMP[36]. To coordinate expression of the T3F with CR attachment to IECs in vivo, we constructed a citrOgen strain in which *mrkABCDF* (excluding *mrkHIJ*) was driven by the promoter of the T3SS effector *map* ($CR_{T3F}$)[37]. The recombinant operon was inserted into the CR *glmS* site (Fig. 6A). We perceived two significant advantages of this strategy, firstly, maximising T3F expression during intimate attachment to host cells and, secondly, to limit the metabolic burden of T3F expression in CR. To confirm accurate gene regulation, we quantified *mrkA* transcript levels in T3SS-suppressing (LB media, static growth) and T3SS-inducing (DMEM, static growth) conditions (Supplementary Fig. 6A)[38]. qRT-PCR analysis demonstrated low *mrkA* transcripts from $CR_{T3F}+pespO$ grown in LB, which increased significantly in DMEM (Fig. 6B); a change also observed for the endogenous *map* gene used as a positive control (Supplementary Fig. 6A). No *mrkA* transcripts were detected in the

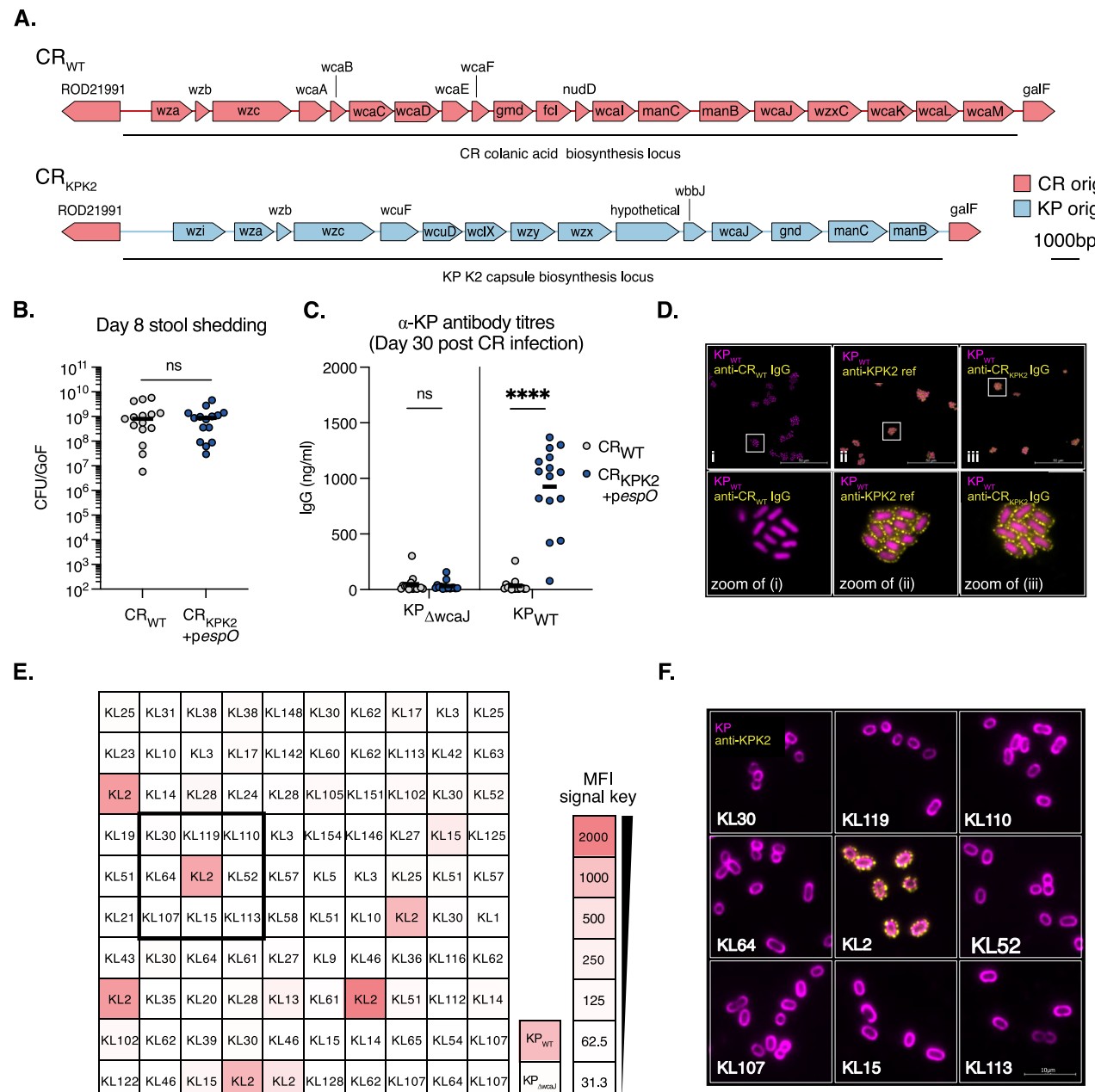

**Fig. 5 | citrOgen derived anti-KP K2 antibodies can be used for serotyping.**
**A** The *cps* in KP_WT is c.22 kb long with 15 open reading frames. This was inserted into CR_WT by substitution of the colanic acid biosynthesis locus generating the strain CR_KPK2. **B** At 8 dpi faecal CR_KPK2+p*espO* levels are similar to CR_WT. Central tendency: median. $n = 15$ mice per group, 3 biological repeats. Two-sided unpaired $T$-test, ns non-significant. GoF, gram of faeces. **C** At 30 dpi blood was taken from mice that cleared infection with CR_WT or CR_KPK2+p*espO*. An ELISA was conducted against heat-killed KP_WT (expressing KP K2) and an isogenic acapsular mutant (KP_ΔwcaJ). CR_WT infected mice had no detectable IgG responses to KP_WT or KP_ΔwcaJ. CR_KPK2+p*espO* infected mice had IgG responses specifically against KP_WT. $n = 15$ mice per group, 3 biological repeats. Central tendency: mean. Two-sided Mann–Whitney test. **D** Sera from CR_WT and CR_KPK2+p*espO*-infected mice were used to stain KP_WT and image by immunofluorescence microscopy. An α-KP K2 pAb was used as a positive control. CR_WT derived sera results in no detectable staining of KP_WT, whereas sera from CR_KPK2+p*espO*-infected mice and the α-KP K2 pAb

phenocopy each other with punctate membrane-associated staining of KP_WT.
**E** Sera from CR_KPK2+p*espO*-infected mice were used to serotype a collection of 100 diverse KP strains with serotypes assigned by genomic analysis. Each box represents one strain with the cps (KL) type described. The boxes are coloured according to the strength of binding by CR_KPK2+p*espO*-infected mice sera, which was assessed by flow cytometry. The median fluorescence intensity (MFI) key is shown in the bottom right together with staining of KP_WT and KP_ΔwcaJ controls. A box, centred on a KL2 isolate is shown. **F** The strains contained within the highlighted box in (**E**) were stained with the reference α-KP K2 pAb and imaged by immunofluorescence microscopy. The KL2 strain in the centre demonstrate punctate membrane staining in keeping with K2 capsule. The other strains (non-KL2) demonstrate no staining. Bacterial cell membranes were counterstained with FM-4-64 for imaging. Images representative of 2 biological repeats. ns non-significant; ****, $p < 0.0001$. Source data and exact $p$-values are provided in the Source Data file.

CR_WT negative control. We then confirmed that CR_T3F formed actin-rich pedestals in vitro and expressed KP T3F by staining with an α-MrkA pAb (Supplementary Fig. 6B). Finally, we tested if T3F conferred CR_T3F the ability to adhere to glass under T3SS inducing conditions.

This revealed a stark increase in the number of adhered CR_T3F observed on glass coverslips compared to CR_WT when both were grown statically in DMEM, indicating functional heterologous T3F expression (Fig. 6C). Whilst this vector was designed to insert and

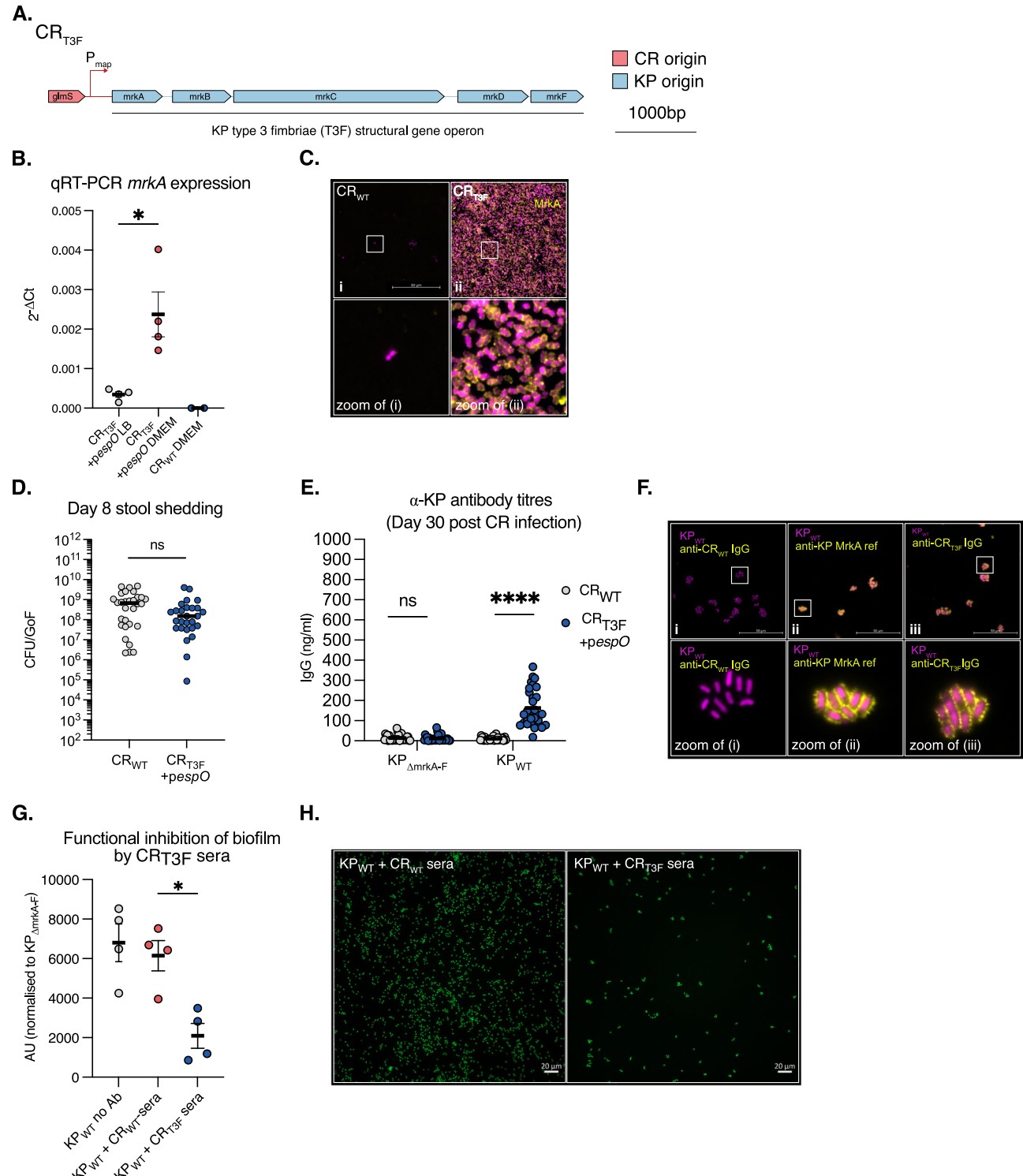

provide stringent control of T3F expression, it can be readily be used for the expression of other operons encoding mono/multimeric protein Ags to facilitate future iterations of this technology.

We started by confirming if EspO overexpression was required in the context of complex protein Ag expression and infected mice with $CR_{T3F}$ and $CR_{WT}$. A similar colonisation defect was evident at 8 dpi to that encountered when KP O1 or KP K2 were expressed in $CR_{WT}$ (Supplementary Fig. 6C). We next used oral gavage to infect CD-1 mice with $CR_{T3F}+pespO$, which colonised similarly to $CR_{WT}$ at 8 dpi (Fig. 6D). We used the sera collected from mice who had cleared the infection for ELISA, with plates coated with heat-killed $KP_{WT}$ and $KP_{\Delta mrkA-F}$, a

deletion mutant in the T3F structural genes. Sera collected from the control $CR_{WT}$-infected mice were not reactive against $KP_{WT}$ or the isogenic mutant (Fig. 6E). Conversely, sera from $CR_{T3F}+pespO$-infected mice bound $KP_{WT}$, with absent binding to $KP_{\Delta mrkA-F}$. This data confirmed the α-KP IgG antibodies were directed towards T3F. Using pooled sera from the $CR_{T3F}+pespO$ infected mice to stain $KP_{WT}$ revealed a binding pattern that matched that of the reference α-MrkA pAb (Fig. 6F). As T3F mediate attachment to abiotic surfaces and biofilm formation, we then tested if the sera from could prevent $KP_{WT}$ attachment. We first pre-adsorbed the sera from $CR_{T3F}+pespO$- and $CR_{WT}$-infected mice against PFA-fixed $CR_{WT}$ to remove α-CR Abs (10×

**Fig. 6 | citrOgen derived anti-KP T3F antibodies block biofilm formation. A** The strain $CR_{T3F}$ expresses the structural *mrkABCDF* genes under the control of the T3SS effector protein Map promoter. **B** *mrkA* mRNA was assessed by qRT-PCR from $CR_{T3F}$ under T3SS suppressing (LB) and inducing (DMEM) conditions. $CR_{WT}$ is also shown as a negative control. DMEM specifically induces *mrkA* transcription in $CR_{T3F}$. 4 biological replicates, central tendency mean, error bars SEM. Two-sided paired *T*-test between LB and DMEM-grown $CR_{T3F}$. **C** $CR_{T3F}$ and $CR_{WT}$ were grown in DMEM on glass coverslip slides. Attachment was assessed at 6 h and bacteria stained for MrkA. A dense lawn of MrkA-positive $CR_{T3F}$ is observed binding to the glass slide, indicating functional KP T3F expression. **D** At 8 dpi, faecal $CR_{T3F}$+*pespO* levels are similar to $CR_{WT}$. Central tendency: median. *n* = 15 mice per group, 3 biological repeats. Two-sided Mann–Whitney test, ns non-significant. GoF, gram of faeces. **E** At 30 dpi with $CR_{WT}$ or $CR_{T3F}$+*pespO* sera was collected. An ELISA was conducted against heat-killed $KP_{WT}$ and an isogenic mutant lacking T3F expression ($KP_{\Delta mrkA-F}$). $CR_{WT}$ infected mice had no detectable IgG responses to $KP_{WT}$ or $KP_{\Delta mrkA-F}$. $CR_{T3F}$ +*pespO* infected mice had specific IgG responses against $KP_{WT}$.

*n* = 15 mice per group, 3 biological repeats. Central tendency: mean. Two-sided Mann–Whitney test. **F** Sera from $CR_{WT}$ and $CR_{T3F}$+*pespO*-infected mice were used to stain $KP_{WT}$. An α-MrkA pAb was used as a positive control. $CR_{WT}$ derived sera results in no detectable staining of $KP_{WT}$. Sera from $CR_{T3F}$+*pespO*-infected mice and α-MrkA pAb staining phenocopy each other. **G** CR preadsorbed sera from $CR_{WT}$ and $CR_{T3F}$+*pespO*-infected mice were preincubated with $KP_{WT}$ or $KP_{\Delta mrkA-F}$. No-antibody was used as an additional control. Following 6 h of growth sfGFP signal was assessed and presented as the $KP_{WT}$ normalised to $KP_{\Delta mrkA-F}$ binding. $CR_{T3F}$+*pespO*-infected mice sera specifically inhibited binding compared to $CR_{WT}$. 4 biological replicates, central tendency: mean, error bars: SEM. Repeated-measures One-way ANOVA corrected with Dunnett's multiple comparisons test. **H** Biofilm formation from (**G**) was assessed by sfGFP fluorescence microscopy. A large reduction in adhered $KP_{WT}$ is observed when $CR_{T3F}$ p*espO*-infected cross adsorbed sera was incubated with $KP_{WT}$ compared to $CR_{WT}$ cross adsorbed sera. ns non-significant; *, $p < 0.5$; ****, $p < 0.0001$. Source data and exact *p*-values are provided in the Source Data file.

24 h adsorption cycles against fresh PFA-fixed $CR_{WT}$) (Supplementary Fig. 6D). This resulted in sera that negligibly bound $CR_{WT}$ (Supplementary Fig. 6E, F). We preincubated these two preadsorbed sera, which had equivalent concentrations of total IgG (Supplementary Fig. 6G), with $KP_{WT}$ expressing sfGFP and measured attachment after static growth at 37 °C. Sera from $CR_{T3F}$+p*espO*-infected mice resulted in threefold reduction in KP attachment compared to sera from $CR_{WT}$-infected mice (Fig. 6G, H). These data demonstrate the generation of α-KP T3F Abs that functionally inhibit surface attachment, the first step in KP biofilm formation.

## Discussion

LPS, CPS and proteins are common bacterial targets where generating specific Abs is desirable. Whilst techniques already exist to generate such target-specific Abs, individually tailored and time-consuming approaches are required to prepare the Ags. Here, instead, we co-opt the mouse pathogen CR for this purpose and combine the steps of Ag preparation and immunisation into a single platform that both expresses heterologous Ags and presents them continuously, in their native conformation, to the host over the entire infection cycle (Fig. 7). This was achieved by recombination-based genome editing in which whole operons were cloned from KP and inserted or substituted into the CR; thus, all genetically encoded components are inherently included, ensuring that Ag assembly takes place in a fashion mirroring the pathogen of choice. It should be noted that, whilst we achieve this for four diverse Ags, they all come from Gram-negative *Enterobacterales*, which will have favoured our approach. Nevertheless, we were able to achieve specific and functional Ab responses for LPS, CPS and a large protein complex. If we consider the latest iteration of the WHO Bacterial Priority Pathogens List[19], then three out of four critical pathogens are Gram-negatives, indicating the clear and present danger posed by this group of bacterial threats. The data we present already supports using citrOgen for bacterial Ag targets in these key organisms. We anticipate that further modifications will be required to synthesise and present Gram-positive bacterial Ags. As well as the ability to present complex polysaccharide structures and multicomponent protein Ags in their native location and configuration, one key advantage of citrOgen is time (Fig. 7). In this work, we have already validated substitution/insertion sites and generated the appropriate genome editing vectors. Changing the inserts encoded by these vectors (to alternative LPS, CPS and protein Ags) is routine and, together with subsequent CR genome editing, can be achieved over a period of days. Whilst we cloned sequences directly from EC and KP DNA, we propose that this technology could also be applied to generate diagnostic or therapeutic Abs in an outbreak scenario. The now routine use of de novo gene synthesis would enable us to engineer strains based on early sequencing data alone. This would allow us to use this platform to generate pAb sera in as little as 28 days- a complete CR infection cycle

in mice. In more controlled settings, we envisage generating citrOgen strains encoding multiple Ags simultaneously. Ethically, this reduces the number of experimental animals for use in research and development and would permit multiple target-specific Ab responses to be obtained following a single infection.

We overcame two major obstacles to develop a viable platform for broader applicability. The first was the potential fitness cost imposed by heterologous Ag expression. Indeed, we observed lower gut colonisation in $CR_{KPO1}$, $CR_{KPK2}$ and $CR_{T3F}$ compared to $CR_{WT}$, a defect that only becomes evident under the pressure of the host-pathogen-microbiome environment. We overcame this obstacle by overexpressing the T3SS effector EspO, which promotes IEC attachment and mucosal colonisation. The second was the metabolic cost to CR of dysregulated heterologous protein expression, which we overcame by placing expression under the control of the endogenous CR promoter for the *map* gene. This restricts heterologous protein expression to T3SS-inducing conditions and promotes presentation at host mucosal surfaces, a strategy that can be used for the expression of any heterologous protein, where expression needs to be tightly controlled and/or where regulatory genes are absent in CR. Further modifications for expression can now easily be implemented, given the rich genetic toolset provided here or that already available for *E. coli*, a close relative of CR, which shares similar genetic tractability.

In the case of KP O1 Ag, we combined citrOgen with subsequent KP lung challenge, which allowed us to quantitively evaluate protection from organ failure conferred by the α-bAb response to this Ag (Fig. 3). Given the current lack of correlates of protection for bacterial infections, we propose that citrOgen strains may represent an efficient tool to evaluate future proposed Ags. Strains engineered to express heterologous Ags (singly or in combination) can be used to evaluate Ab responses following a CR infection cycle. In the same cohort of mice, subsequent pathogenic challenge can then evaluate the protection conferred by the response. Expanding the translational applicability of current priority bacterial infection models would significantly enhance the data we could achieve in this approach.

CR infections are host-restricted to mice, conferring platform safety and preventing infection risk to humans. This is especially important as future citrOgen strains could include additional virulence factors as Ags for Ab generation. We currently use the citrOgen platform in conventional mice, generating murine immunoglobulins. Advances in molecular genetics have enabled the generation of "humanised" Abs initially raised in non-human species and could be applied to the current iteration of the platform[39]. This improves IgG tolerability and durability when administered to human patients, largely by reducing the immunogenicity provoked by non-human Ig regions. Furthermore, extensive work in the field of mouse genetics has enabled the generation of fully human Abs in transgenic animals[40]. Combining these transgenic mice with citrOgen strain infection would

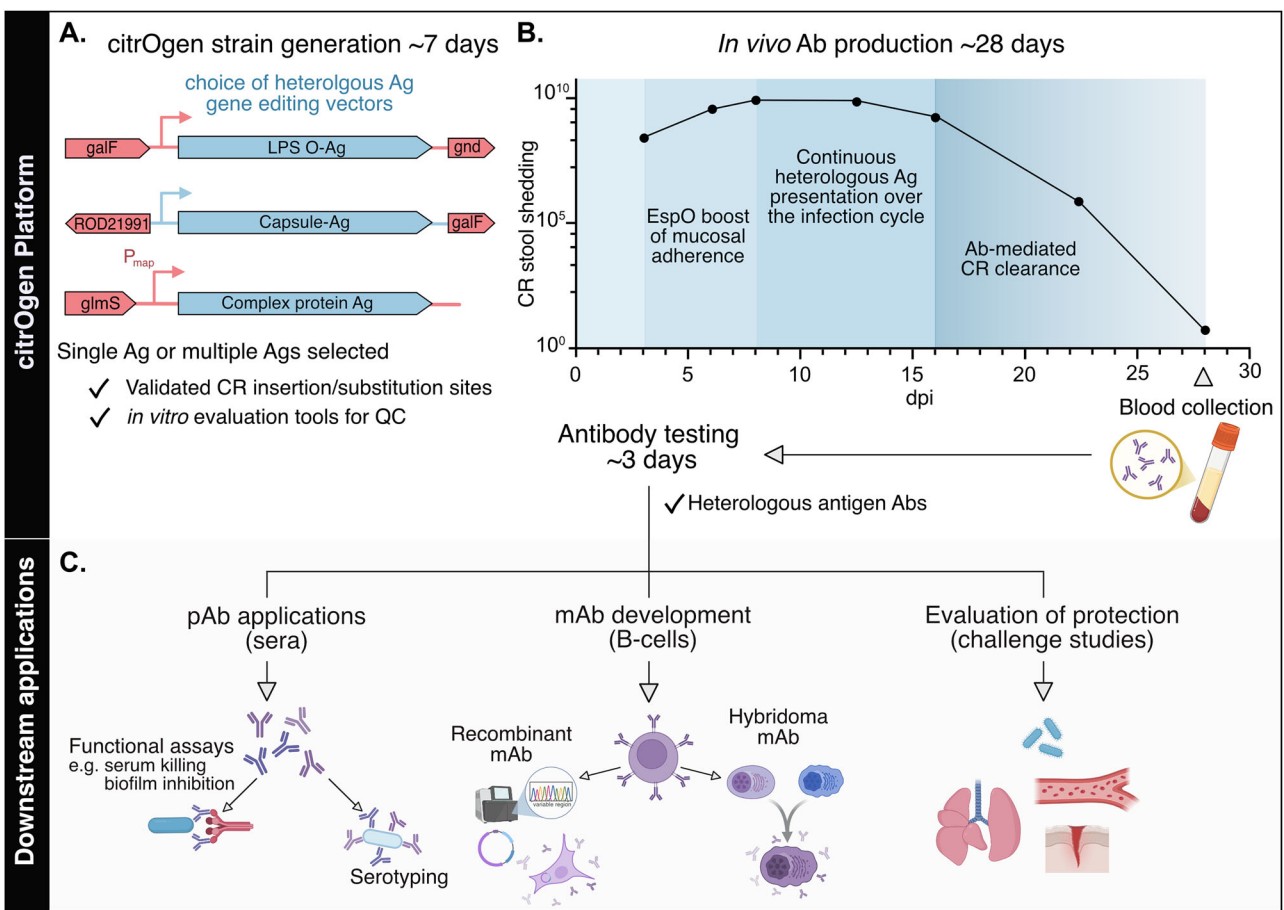

**Fig. 7 | The citrOgen platform as a versatile, quick and cost-effective method of Ab production and multiple downstream applications. A citrOgen strain generation.** Selected Ag sequences are incorporated into the appropriate gene editing vectors designed for optimal chromosomal expression of heterologous O-Ag, capsule or complex protein Ags, the latter under the map promoter to ensure timely expression during mucosal infection. p*espO* is then transformed to finalise the new citrOgen strain. QC checks include assessment of heterologous expression and pedestal formation in vitro (as shown in Supp Figs. 1A, 4A and 5B) before proceeding to in vivo Ab production. **B In vivo Ab production**. The CR infection cycle and immune responses are co-opted by the citrOgen platform: (i) EspO-mediated boosting of CR attachment to intestinal epithelial cells, (ii) continuous exposure of immune cells to the heterologous Ags, (iii) B cell affinity maturation results in Abs, primarily IgG, which mediate CR clearance by 28 dpi and can be obtained as pAb sera from blood. **C Downstream applications**. Following citrOgen infection/immunisation, the functionality of Abs can be assessed in vitro or by in vivo challenge studies, the pAb sera can be used for serotyping and the B-cells can be used for mAb production, either by using hybridoma technology to immortalise single-cell cloned B-cells or by expressing heterologous Ag-specific Ab sequences in cell lines to obtain recombinant mAb. Ab Antibodies, mAb mono-clonal antibodies, pAb polyclonal antibodies, Ag Antigen, CR *Citrobacter rodentium*. Created using BioRender. Frankel, G. (2025) https://BioRender.com/h0xs2kn.

result in human Ig induction in one CR infection cycle (Fig. 7). Existing downstream workflows, such as hybridoma generation or IgG loci subcloning, are already well defined and employed to produce mAbs from B-cells derived from these mice. Whilst our antibiotic treatment options are rapidly diminishing, the application of this technology may alleviate future infectious diseases deaths by generating new Ab-based diagnostic tools and therapeutics.

## Methods

### Ethical statement

All animal work took place at Imperial College London (Association for Assessment and Accreditation of Laboratory Animal Care accredited unit) under the auspices of the Animals (Scientific Procedures) Act (UK) 1986 (PP7392693). Work was approved locally by the institutional ethics committee (AWERB).

### Bacterial culture and antibiotics

Bacteria were routinely cultured in LB Broth (Miller) (84649.05000, VWR, UK). Saturated (overnight, 18 h) cultures took place by inoculating broth and incubation at 37 °C and shaking at 200 rpm.

Antibiotics were added to LB broth or LB broth solidified with agar (1.10283.0500, VWR, UK) at the following concentrations:
Gentamicin 10 mcg/ml (cloning) (G1272, Merck, UK)
Streptomycin 50 mcg/ml (cloning) (S6501, Merck, UK)
Kanamycin 50 mcg/ml (cloning) Merck, UK)
Nalidixic acid 50 mcg/ml (selecting CR and CFU enumeration from in vivo sample) (N4382, Merck, UK)
Rifampicin 50 mcg/ml (selecting KP and CFU enumeration from in vivo sample) (R3501, Merck, UK)

### Cloning

High-fidelity polymerase chain reaction was conducted using Phanta Flash (2× Phanta Flash Master Mix (Dye Plus), Vazyme, China). Screening and colony polymerase chain reaction was conducted using Rapid Taq (2× Rapid Taq Master Mix, Vazyme, China).

PCR purification (Monarch PCR & DNA Cleanup Kit (5 μg), NEB) and plasmid extraction were conducted with kit-based methods.

All mutagenesis vectors were generated by PCR products ligation via Gibson Assembly (NEB, UK) according to the manufacturer's guidance.

## Genome editing

The whole genome references sequences used for CR strain ICC168 (Nalidixic acid-sensitive version of ICC169) is FN543502. The whole genome reference sequence used for ICC8001 is the closely related ATCC43186-derivitive KPPR1 CP009208.1 [https://www.ncbi.nlm.nih.gov/nuccore/CP009208.1]. A list of strains generated in this work can be found in Supplementary Data 1.

Two methods of genome editing were conducted in this work. Insertions at the CR 3' *glmS* site were conducted using Tn7 mutagenesis with methods published in Choi et al. Nat Methods 2005 Jun;2(6):443-8[24]. In this work Tn7 vector (100 ng) and pTNS2 (100 ng) plasmid were co-transformed into room temperature competent CR. Briefly, a single colony was grown overnight in LB, diluted 1:100 into fresh LB, and incubated at 37 °C with shaking at 900 rpm until mid-log phase (~ 2 h). Cells are pelleted and washed twice with autoclaved Milli-Q water, then resuspended a third time for a final wash and concentrated into 30 µL of sterile water. Plasmid DNA was then added directly to this suspension and electroporation was performed immediately, with no incubation step. Bacterial cells were recovered in SOC (Super Optimal medium with catabolic repressor) media and plated in LB agar supplemented with the corresponding antibiotic. Seamless, markerless genome editing at other sites was conducted by homologous recombination based on the pSEVA612S system (https://seva-plasmids.com/). To improve the efficiency of pSEVA612S for Gibson Assembly, we further modified this plasmid by reversing the sequence of the 3' ISceI endonuclease digestion site, generating pSEVA612S(R). This modification improves efficiency by reducing homology-based empty vector recombination. First step recombination was facilitated by triparental mating with donor strains (carrying mutagenesis vectors, Supplementary Data 2) overlayed with the *E. coli* helper strain carrying pRK2013. This was followed by overlay with either CR or KP recipients transformed with pACBSR-SmR. Following selection of first step recombinant, second step recombinants were induced by arabinose induction (0.5% w/v) in LB. Mutants were screen and confirmed by PCR. These protocols have been published in detail for both CR and KP[12,25].

All mutagenesis vectors are summarised in Supplementary Data 2 and are publicly available at https://doi.org/10.6084/m9.figshare.c.7869659.v1[41]. In the case where more than one vector was used these are listed against the respective strain. To generate CR_KPK2 we initially cloned a construct (pSEVA612S(R)_CRColanic::K2cpsΔwbbJ_ORF) without the genes wbbJ and an annotated open reading frame of unknown function[42]. We subsequently inserted these genes into our final CR_KPK2 strain (pSEVA612S(R)_ΔwbbJ::Ins_wbbJ_ORF) as new data become available[43] and included the native KP promoter (pSEVA612S(R)_K2:col_Pvariant_Porf2).

**Transformation of pACYC184_espO was conducted using room temperature competent cells according to our previously published protocol25 and as outlined above.** All primers are appended to GenBank files and publicly available at https://doi.org/10.6084/m9.figshare.c.7869659[41]. Molecular cloning primers were designed using Benchling (https://www.benchling.com/).

## Sequencing

Sanger sequencing of PCR products and plasmids was conducted by Eurofins Genomics.

## In silico serotyping

In silico serotyping of CR was conducted using SerotypeFinder 2.0 (https://cge.food.dtu.dk/services/SerotypeFinder/) on genome FN543502.

## Crude LPS preparation and Silver stain

1 ml of overnight bacterial cultures normalised to $OD_{600}$ of 1.2 were pelleted by centrifugation at $10,000 \times g$ and resuspended in 200 µl Laemmli buffer, followed by boiling at 100 °C for 5 min. Proteinase K was added to a final concentration of 1 mg/ml, and samples were incubated at 60 °C for 2 h. Subsequently, 5% β-mercaptoethanol was added, and samples were incubated for an additional 5 min. Crude LPS samples were separated by SDS-polyacrylamide gel electrophoresis (PAGE) using a 12% gel, and the resulting gels were fixed in 5% acetic acid/40% isopropanol overnight. Silver staining was then performed using the SilverQuest™ Silver Staining Kit (LC6070, Thermo Fisher Scientific) according to the manufacturer's protocol. All steps were carried out at room temperature with gentle agitation. Images were acquired on a ChemiDoc XRS+ and visualised using Image Lab software (Bio-Rad Laboratories).

## LPS extraction and Western blotting

LPS was extracted from 3OD units of saturated overnight cultures using a modified phenol method kit (iNtRON Biotechnology, Korea) according to the manufacturer's guidance with no modification.

Samples were mixed with 5× Laemmli buffer and 5% 2-mercaptoethanol and separated by SDS−PAGE using a 12% acrylamide gel and tris-glycine buffer system. and transferred to polyvinylidene difluoride membranes (#1620177, Bio-Rad Laboratories) using a TransBlot semi-dry electrophoretic transfer machine (Bio-Rad). Membranes were blocked for 2 h at room temperature (RT) in 10% fat-free milk PBS−0.1% Tween 20 (PBS-T) and incubated overnight at 4 °C with 1:500 anti-O152 or 1:2000 anti-KP O1 (Supplementary Data 3) in PBS-T with 5% BSA (A3059-50G, Sigma). Secondary antibodies (Supplementary Data 3) were added diluted 1:10000 in 5% fat-free milk in PBST and incubated for 1 h at RT. Immunoblots were developed with Clarity Western enhanced chemiluminescence (ECL, Bio-Rad Laboratories) and images were acquired on a ChemiDoc XRS+ and visualised using Image Lab software (Bio-Rad Laboratories).

## In vitro CR_KPO1, CR_KP K2 and CR_T3F infections of fibroblasts

Murine Swiss 3T3 fibroblasts (ATCC® CCL-92™) were maintained in DMEM containing 4500 mg $l^{-1}$ glucose (D6546, Sigma) and supplemented with 10% fetal calf serum (F9665, Sigma) and 2 mM glutaMAX (#35050061, ThermoFisher Scientific) at 37 °C in 5% (v/v) $CO_2$ and seeded on glass coverslips in 24-well plates (density $2 \times 10^5$ cells/well) for immunofluorescence. Indicated strains of CR expressing mCherry from a pUltra plasmid[44] were grown to saturation in LB supplemented with gentamicin at 37 °C with 200 rpm agitation for 8 h before subculturing 1:500 for 16 h in LB or DMEM with 1000 mg $l^{-1}$ glucose (# D5546, Sigma) at 37 °C, 5% (v/v) $CO_2$, static. Cells were infected with the static culture at a multiplicity of infection of 100, confirmed by retrospective plating. The plate was centrifuged at $700 \times g$ for 5 min at RT to synchronize infection and incubated for 4 h at 37 °C in a 5% $CO_2$ atmosphere. The infection was stopped with three washes with sterile phosphate-buffered saline (PBS) and cells were fixed in PBS with 4% (w/v) paraformaldehyde for 15 min at RT and washed three times in PBS. Cells were then permeabilized in 0.1% (v/v) of Triton ×100 in PBS for 10 min, incubated in 50 mM $NH_4Cl$ in PBS as a quenching agent for 10 min, washed in PBS, and blocked in 3% BSA in PBS for 1 h at RT before staining with one of the following primary antibodies in 1% BSA-PBS for 1 h at RT: anti-KPO1 C13 (1:200), anti-mrkA (1:100) or anti-KP K2 (1:100) (Supplementary Data 3). Coverslips were washed in PBS and incubated with Alexafluor488-conjugated secondary antibodies (1:200), Tetramethylrhodamine (TRITC)−conjugated phalloidin (1:500) and 4′,6-diamidino-2-phenylindole (DAPI) (1:1000) to label the heterologously-expressed antigen, F-actin and DNA, respectively (Supplementary Data 3) in 1% BSA-PBS for 1 h. Coverslips were washed in PBS and once in water before being mounted using Gold Pro-Long Anti-fade medium (#P36930, ThermoFisher Scientific) and left to cure overnight at RT in the dark. Immunofluorescence images were acquired using a Zeiss AxioImager Z1 microscope and processed using ZEN3.1 software (ZEN lite, Zeiss).

## Reverse transcription quantitative PCR (RT-qPCR)

The indicated strains of CR were grown to saturation overnight in LB supplemented with the corresponding antibiotics. 0.5 ml of the overnight culture were washed once in PBS, incubated in RNAprotect Bacteria reagent (#76506, Qiagen) for 10 min at RT and centrifuged at $3000 \times g$ for 5 min. The bacterial pellet was digested in 100 µl TE buffer (30 mM Tris·HCl, 1 mM EDTA, pH 8.0) with 15 mg/ml Lysozyme (L6876, Sigma-Aldrich) and 20 µl Proteinase K (#19131, Qiagen) for 10 min according to manufacturer's guidelines. RNA was isolated using the RNeasy minikit (Qiagen) following the manufacturer's instructions, and RNA concentration and purity was determined using a Nanodrop 2000 spectrophotometer (ThermoFisher Scientific). 1 µg of RNA was treated with RNase-free DNase (#M6101) for 1 h at 37 °C and cDNA was then synthesized using a Moloney murine leukemia virus (M- reverse transcriptase (M-MLV, #M1705) with random primers (#C1181) following the manufacturer´s instructions (all from Promega). To confirm the absence of undigested DNA, a reaction mixture without the M-MLV reverse transcriptase was also included (NRT). qPCR was performed using the Power Up SYBR Green master mix (#A25778, ThermoFisher Scientific) and the relevant primers, whose primer efficiency had been evaluated previously and determined to be 95–105% (Supplementary Data 4). The assay was run on a Quantstudio 1 System (Applied Biosystems, ThermoFisher Scientific) and results were analysed using the QuantStudio Design & Analysis Software v1.5.2 (Applied Biosystems, ThermoFisher Scientific). The relative expression level of each mRNA was evaluated using the 2-ΔCtmethod, where Ct is the threshold cycle and 16S rRNA is used as the control.

## O25 antisera agglutination

Heat-killed agglutination was performed on the indicated strains after overnight growth in LB. The anti-O25 sera (85022, SSI Diagnostica) was used according to the manufacturer's protocol. Images were acquired on a ChemiDoc XRS+ and visualised using Image Lab software (Bio-Rad Laboratories).

## Whole bacterial cell ELISA

To prepare heat-killed KP the indicated strains were grown overnight in LB at 37 °C at 200 rpm. The culture was then normalised to an optical density of 1 at 600 nm in sterile PBS, heat inactivated in the presence of EDTA-free protease inhibitor cocktail (#4693159001, Sigma) at 60 °C for 1 h and stored at 4 °C until use (short-term). ELISA Flat-Bottom Maxisorp Immuno 96- Well Plates (#10547781, Fisher Scientific) were incubated with 100 µL/well heat-killed CR for our serum samples or with 100 µL/well goat anti-mouse IgG antibody for the standards (2 µg/µL) (Supplementary Data 3) overnight at 4 °C. Plates were washed three times with PBS-T, blocked for 1 h at RT with 5% BSA-PBS-T, and incubated overnight at 4 °C with serum samples or serially diluted mouse IgG (Supplementary Data 3) as standard. Plates were washed as described above and incubated with horseradish peroxidase (HRP)-conjugated goat anti-mouse IgG antibody (Supplementary Data 3) for 1 h at room temperature. Plates were washed and developed by the addition of 100 µL 3,3′,5,5′-tetramethylbenzidine (TMB; Thermo Fisher Scientific) for 5–15 min before stopping the reaction with 2 N $H_3PO_4$. The absorbances at 450 and 540 nm were measured in a plate reader (FLUOstar Omega; BMG Labtech), and anti-KP IgG levels were quantified in samples by comparison to a mouse IgG standard curve.

## Determination of O-antigen-binding specificity

$KP_{WT}$ expressing mCherry and $KP_{\Delta rfb}$ expressing sfGFP were grown as saturated overnight cultures in LB supplemented with gentamicin. On the following day, both strains were mixed 1:1 (v/v) after confirming the had grown to the same optical density (600 nm) and transferred to a V-bottom 96-well plate (Greiner Bio-one) for staining. Single strain controls were also included to ensure correct gating during flow cytometry analysis. Samples were first blocked in 5% BSA/PBS for 1 h at RT shaking, centrifuged at $3200 \times g$, 4 °C, 10 min and then incubated overnight with sera from $CR_{WT}$ or $CR_{KPO1}$+pespO-infected mice diluted 1in100 (v/v) in 1% BSA/PBS. Bacterial samples were centrifuged, washed twice in cold PBS and resuspended in 1% BSA/PBS containing AlexaFluor647-conjugated anti-mouse IgG (1:200, Supplementary Data 3) for 1 h at RT shaking. Cells were washed twice in cold PBS, fixed in 1% PFA/PBS and kept in the dark until flow cytometry analysis. Samples were acquired (minimum of 100000 bacterial cells per sample) on a Cytek Amnis CellStream and data was analysed using FlowJo v10.10.0 (gating strategy shown in Supplementary Fig. 3A).

## K2 serotyping

The KP MRSN Diversity Panel (NR-55604) was obtained through BEI Resources, NIAID, NIH[33]. The 100 strains from the MRSN collection, $KP_{WT}$ and $KP_{\Delta wcaJ}$ (positive and negative controls respectively) were grown overnight in 1 mL of LB in deep well 96-well plates (VWR). 100 µL of each overnight was transferred to a V-bottom plate and used for staining. Plates were centrifuged at $3200 \times g$ for 10 min at 4 °C, resuspended in 5% BSA/PBS and blocked for 1 h at RT shaking using a microplate mixer. Bacterial samples were then centrifuged and stained overnight at 4 °C shaking using sera from $CR_{KPK2}$+pespO-infected mice diluted 1in100 (v/v) in 1% BSA/PBS. On the following day, bacterial samples were washed two times in cold PBS and stained for 1 h at RT, shaking with AlexaFluor488-conjugated anti-mouse IgG (1:300; Supplementary Data 3). Cells were washed twice in cold PBS, fixed in 1% PFA/PBS and kept in the dark until flow cytometry analysis. Samples were acquired on a Cytek Amnis CellStream (minimum of 50000 bacterial cells per sample) and data was analysed using FlowJo v10.10.0.

## Bacterial cell immunofluorescence microscopy (adherent protocol and suspension protocol)

We used two protocols for immunofluorescent staining of KP. All images were obtained using a Zeiss AxioVision Z1 microscope with a Hamamatsu microscope camera and processed using Zen 2.3 Blue Version (Carl Zeiss MicroImaging GmbH, Germany).

## Adherent protocol (all ICC8001 staining)

mCherry expressing ICC8001 was grown as a saturated overnight culture, which was sub-cultured (1:1000 v/v) into M9 media (M9 salts M6030, Sigma, UK) supplemented with 0.4% glucose w/v (G8270, Sigma, UK). 400 µL was added into each well for imaging in a coverslip slide (µ-Slide 8 Well high Glass Bottom, ibidi). Slides were grown statically at 37 °C for 4 h and cells were washed in phosphate-buffered saline before staining. Primary staining (sera, antibodies) took place overnight at 4 °C in 0.22 µm filtered 3% bovine serum albumin (A3059, Sigma, UK)/ PBS on a rocking platform. Primary antibody was removed, and slides were washed 3× in PBS followed by incubation with fluorescently labelled secondary antibodies (Supplementary Data 3) in 3%BSA/PBS at room temperature for 1 h on a rocking platform. Slides were washed 3 times before imaging.

## Suspension protocol (MRSN collection, Fig. 4D)

MRSN collection strains were grown as saturated overnight cultures. 100 µL of overnight was used for staining. Following centrifugation ($5000 \times g$, 5 min), cells were washed 3 times in PBS. Primary staining (K2 antisera) took place overnight at 4 °C in 3%BSA/PBS in 400 µL volume with agitation. Cells were washed by sequential spins ($5000 \times g$, 5 min) and resuspension in PBS (3 washes). Secondary antibody (Supplementary Data 3) was added for 45 min at room temperature in a 100 µL volume with agitation. Cells were washed by sequential spins ($5000 \times g$, 5 min) and resuspension in PBS (3 washes). Cell membranes were then labelled with FM® 4–64 Dye (T13320, ThermoFisher Scientific, UK) at a final concentration of 5 µg/ml in 100 µl of PBS for 10 min at room temperature in the dark. Cells were

washed once and resuspended in 100 μL of fresh PBS before imaging. To facilitate high-resolution imaging of the bacterial cells in suspension, cells were immobilised using agarose pads. A thin layer of ~1% low-melt agarose was cast between glass slides and allowed to solidify before being sectioned into small squares. A droplet of bacterial culture was then applied to a coverslip, overlaid with the agarose pad and a second coverslip, effectively immobilising the cells for imaging.

### $CR_{T3F}+pespO$ glass binding assay

mCherry expressing $CR_{T3F}+pespO$ and $CR_{WT}$ were grown as saturated overnight cultures and sub-cultured (1:100) into DMEM low glucose (D5546, Merck, UK) into glass coverslip slides (μ-Slide 8 Well high Glass Bottom, ibidi). Slides were grown statically at 37 °C for 6 h and cells were washed in phosphate-buffered saline before fixing in 4% paraformaldehyde/PBS (R1026, Agar Scientific, UK) for 20 min at room temperature. Cells were stained for MrkA according to the adherent protocol outlined above.

### Pre-adsorption of CR sera

Serum samples from mice infected with $CR_{T3F}+pespO$ with anti-KP antibodies above 200 ng/mL were pooled; sera from $CR_{WT}$-infected mice were also pooled separately as a control. To inactivate complement, pooled serum samples were heated at 55 °C for 1 h and then were pre-adsorbed onto $CR_{WT}$ expressing sfGFP to remove CR-specific antibodies, thus enriching for anti-KP antibodies. This was done by serially incubating the pooled sera samples, both from $CR_{WT}$- and $CR_{T3F}+pespO$-infected mice, with ~$5 \times 10^8$ $CR_{WT}$ for 24 h at 4 °C on a rotating wheel for 10 rounds of pre-adsorption. After each 24h-incubation, the serum samples with CRWT were centrifuged at $3000 \times g$ for 5 min to pellet the bacteria, which were kept at 4 °C until flow cytometry analysis; the supernatant containing the antibodies was subsequently incubated with a fresh batch of ~$5 \times 10^8$ $CR_{WT}$ expressing sfGFP. At the end of the 8 rounds, sfGFP-expressing bacterial samples from each round of pre-adsorption with sera from $CR_{WT}$- and $CR_{T3F}+pespO$-infected mice were stained with AF555-conjugated anti-mouse IgG (1:400; Supplementary Data 3) in 2% BSA-PBS for 30 min and analysed using a Cytek Amnis CellStream. Obtained data was subsequently analysed using FlowJo v10.10.0.

### Dot blot

Bacterial cultures were grown overnight in LB (CR) or M9 supplemented with glucose (KP), normalised to an OD600 of 1 and centrifuged to pellet the cells. The pellet was resuspended in 1 ml PBS and equal volumes of bacteria were then spotted onto a nitrocellulose membrane (#88018, ThermoFisher Scientific) and allowed to air dry. The membrane was blocked at RT for 2 h in 5% milk-PBS-T and incubated overnight at 4 °C with the indicated sera diluted in 5% BSA-PBS-T to a concentration of 2 μg/ml of total IgG. The primary was washed three times in PBS-T and a secondary HRP-conjugated anti-mouse antibody (Supplementary Data 3) was added in 5% milk-PBS-T, incubated for one hour at RT, and washed three times. Detection was carried out using ECL substrate (Bio-Rad) and images were acquired on a ChemiDoc XRS+ and visualised using Image Lab software (Bio-Rad Laboratories).

### Fluorescent biofilm formation assay

$KP_{WT}$ and $KP_{\Delta mrkA-F}$ expressing sfGFP were grown overnight in M9 media and diluted into fresh M9 1:500 (v/v) and preincubated for 30 min with no sera or with ~50 μg/mL IgG (1:100 dilution) of the pre-adsorbed sera from either $CR_{WT}$- and $CR_{T3F}+pespO$-preinfected mice at RT on a rotating wheel. 100 μl of each sample in triplicate were transferred into wells in a black/clear-bottom 96-well tissue-culture treated plate (#353219, SLS) and incubated statically at 37 °C for 6 h. Samples were then washed 5 times with sterile PBS and bottom optic

fluorescence readings were taken at 485/520 (Ex/Em) using a FLUOstar Omega plate reader. Fluorescence values were normalised to those obtained with the $KP_{\Delta mrkA-F}$ strain, which does not attach, to remove any non-specific fluorescence and reduce batch-to-batch variation.

### Animal work

**Animals.** Female CD-1 and C57BL/6 mice were purchased from Charles River, UK. Mice were purchased at $29-31 \times g$ weight (5–7 weeks) (CD-1 mice) and $18-20 \times g$ weight (6–8 weeks) (C57BL/6 mice). For each experiment, mice were randomly assigned to experimental groups. Investigators were not blind to the allocation. C57BL/6 were used for the data shown in Supplementary Fig. 2C, all experiments directly related to the citrOgen platform were performed in CD-1.

### Husbandry

Animal technicians within the facility received mice and blindly separated them into groups of 5 into high-efficiency particulate air–filtered cages with sterile bedding. Bedding was changed weekly, and enrichment supplied. Mice were housed under a 12/12-h light/dark cycle with access to food and water *ad libitum*. Researchers (JSG and JLCW) conducting the work in this study were not involved in selection of animals into groups as this was undertaken by technical staff prior to starting experiments. Animals were identified by ear notching (conducted under restraint with no anaesthesia).

### CR infection by oral gavage

For infections, the indicated CR strains were grown overnight in 15 ml of LB (50 ml capped Falcon tubes) at 37 °C and 200 rpm. The next day, the tubes were centrifuged, and the pellet was resuspended in 1.5 ml of sterile PBS. Mice were infected by oral gavage with 200 μl of this solution ($1 \times 10^9$ CFU confirmed by viable counts as previously described[13]). CR faecal shedding was determined by collecting stool samples at the specified days post-infection (DPI), homogenising the samples in PBS and plating serial dilutions onto LB agar containing nalidixic acid. Plates were incubated at 37 °C overnight and colonies were counted on the following day.

Mice were culled at 6 (C57BL/6) or 8 dpi (CD-1) and colons were extracted for downstream processing. When characterising antibody responses, mice were followed until clearance of CR infection and then culled to obtain sera or challenged with KP infection.

### Colonic histology processing and immunostaining

At 8 DPI, the colon was extracted, and the distal 0.5 cm was fixed in 4% PFA for 2 h and then kept in 70% ethanol until processing. Fixed tissues were processed, paraffin embedded and sectioned at 5 μm. For immunofluorescence, sections were dewaxed by submersion in HistoClear II solution (#101412-884, VWR) twice for 10 min, 100% ethanol twice for 10 min, 95% ethanol twice for 10 min, 80% ethanol once for 3 min, and PBS-0.1% Tween 20-0.1% saponin (PBS-TS) twice for 3 min. Subsequently, sections were heated for 30 min in demasking solution (0.3% trisodium citrate-0.05% Tween 20 in distilled H2O). Once cooled, slides were first blocked in PBS-TS supplemented with 10% normal donkey serum (NDS, D9663, Sigma) for a minimum of 1 h in a humid chamber, before being incubated with anti-CR antibody (1:50)[45] and anti-KP O1 (1:100) diluted in PBS-TS with 10% NDS overnight at 4 °C. Slides were rinsed twice for 10 min each time in PBS-TS, followed by incubation with Alexafluor488-conjugated secondary antibodies (1:100) and DAPI (1:1000) (Supplementary Data 3). Washing steps were repeated before slides were mounted with ProLong Gold antifade mountant. Images were acquired using a Zeiss AxioVision Z1 microscope using a Hamamatsu microscope camera and processed using ZEN3.1 software (ZEN lite, Carl Zeiss MicroImaging GmbH, Germany).

## cIEC isolation and flow cytometry analysis

Colonic intestinal epithelial cells (cIECs) from mock-infected and CR-infected C57BL/6 mice were isolated from 2-cm distal colonic tissue samples as previously described[46]. Briefly, the 2-cm colonic tissue sample was opened longitudinally and washed in 1× Hanks' balanced salt solution (HBSS) without Mg and Ca (# 14185045, ThermoFisher Scientific) and incubated at 37 °C, 200 rpm for 45 min in IEC dissociation buffer (1× HBSS, 10 mM HEPES, 1 mM EDTA, and 5 μL/mL 2-β-mercaptoethanol). The remaining tissue was removed, and detached cells were collected by centrifugation ($2100 \times g$ for 10 min), followed by two DPBS (#D8537, Sigma) washes at 4 °C. Cells were then incubated 5 min at RT with 50 μg/mL DNase I (#11284932001, Sigma) to digest extracellular DNA and passed through a 70-μm cell strainer on ice to ensure a single-cell suspension, pooling samples from the same group and rinsing with 20 mL of cold DPBS. All steps from here onwards were performed on ice/at 4 °C to maintain cell viability. cIECs were counted using trypan blue to exclude dead cells and $\sim 1 \times 10^7$ live cells were centrifuged, resuspended in fluorescence-activated cell sorter (FACS) buffer (5% FBS, 2 mM EDTA in DPBS) supplemented with Fc block (#130-092-575, Miltenyi Biotec) and incubated for 10 min. Samples were then incubated with FACS buffer containing anti-CR (1:1000) for 20 min, washed twice in and then incubated with EpCAM-APC (1:100) and phycoerythrin (PE) anti-rabbit IgG (Supplementary Data 3) for 40 min, washed, and fixed in 1% paraformaldehyde in PBS prior to acquisition on a BD LSRFortessa cell analyser with BD FACS-Diva v9.0 Software (BD Biosciences). Data were analysed using FlowJo v10.10.0 and the gating strategy is shown in Supplementary Fig. 2A.

## Tail vein bleeds and serum separation

Small volume bleeds (< 100 μL) were conducted on conscious, restrained mice using a Tail Vein Injection platform (Hugeiron, MouseSerum, China). Blood was taken from the lateral tail vein by puncture with a 25G needle and collected via pipette into a paediatric serum collection tube (Microvette® 100 Serum Cat, Starstedt). Following 1 h of clotting at RT, sera was separated by centrifugation at $20,000 \times g$ for 3 min. Separated sera was stored at −80 °C until use.

## KP pulmonary infections

Anaesthesia (ketamine 80 mg/kg and medetomidine 0.8 mg/kg) was delivered by intraperitoneal injection. Body temperature was maintained with a heat mat throughout until recovery or during the intubation procedure itself.

Intubation was achieved with a 20G cannula (BD Insyte, BD) using the commercially supplied intubation kit (Endotracheal Mouse Intubation Kit, ETI-MSE, Kent Scientific). 50ul of inoculum (diluted in phosphate-buffered saline) was delivered into the hub of the 20G cannula and drawn into the lungs by spontaneous ventilation.

Recovery took place at 32 °C in a forced air heating cabinet. At 15 min post anaesthesia induction medetomidine was reversed by the administration of 0.8 mg/kg atipamezole, delivered by subcutaneous injection into the neck scruff.

The total volume of administered drugs was designed to deliver 20 mg/kg of total fluids to animals, a human paediatric resuscitation dose.

Mice were returned to their home, individual ventilated cages, after they were spontaneously moving and checked at regular intervals post-infection.

## Cardiac puncture

Mice were weighed at end-point (after clearance for CR infected animals and 72 hpi for KP-challenged animals).

Anaesthesia (ketamine 100 mg/kg and medetomidine 1 mg/kg) was delivered by intraperitoneal injection. Mice were checked and following the absence of pedal reflexes, placed in dorsal recumbency. Paws were taped to a board and scissors used to remove the skin over

the xiphisternum. Blood was collected from the right ventricle (closed approach) with a 25G needle inserted at 30° to the skin surface until blood was freely aspirated into a 1 ml syringe.

When blood was obtained for determination of IgG titres (i.e. mice infected with indicated CR strains expressing heterologous antigens), the 25G needle was removed (to prevent haemolysis) and the blood was transferred to a serum collection tube containing a clot activator (BD Microtainer SST with separating gel, BD). Blood in the serum tube was allowed to clot for 1 h at RT before centrifugation ($20,000 \times g$) for 3 min. The sera was divided into aliquots and stored at −80 °C until use.

When blood was obtained from KP challenged mice, 20 μL of whole blood was taken for CFU enumeration by plating tenfold serial dilutions (in PBS supplemented with 1 mmol EDTA to prevent clotting) on LB Agar plates containing rifampicin 50 μg/ml. The 25 G needle was again removed and approximately 50–100 μl placed immediately into a tube coated with EDTA for anticoagulation (Microvette® 100 EDTAK3E, 100 μl, Sarstedt) for haematological analysis. The remaining blood (approximately 800 μl) was allowed to clot for 1 h at room temperature in a serum collection tube as described above, before centrifugation ($20,000 \times g$) for 3 min. The sera was divided into aliquots and stored at −80 °C until use.

## Lung homogenate processing

Post-mortem lungs were removed and placed in 3 ml of phosphate-buffered saline in a gentleMACS C-tube (Miltenyi Biotec). These were homogenised on a gentleMACS Octo Dissociator (Miltenyi Biotec) using the program m_lung_2. The program was run twice in succession to achieve uniform homogenisation.

30 μL of lung homogenate was taken for CFU enumeration by plating tenfold serial dilutions on LB Agar plates containing rifampicin 50 μg/ml.

1 ml of lung homogenate was transferred into a 1.5 ml microcentrifuge tube and spun at $10,000 \times g$ for 5 min at 4 °C to pellet out particulate matter. The supernatant was stored −80 °C for future use.

## HM5 analysis of EDTA anticoagulated whole blood

Anticoagulated blood was run on the HM5 Vetscan analyser according to the manufacturer's instruction.

Outcome variables of infection were assessed using the kits summarised in Supp Table 5 according to the manufacturer's guidelines with the following specifications.

**Capture ELISAs.** ELISAs were performed using ELISA Clear Flat-Bottom Maxisorp Immuno 96- Well Plates (#10547781, Fisher Scientific). Samples were added to coated and blocked plates and incubated overnight at 4° for all ELISAs. The range of sample dilutions used for each ELISA is indicated in Supplementary Data 5; serial dilutions in the corresponding assay diluent were performed when samples needed to be diluted further than 1in50. After addition of the TMB substrate solution (#00-4201-56, ThermoFisher Scientific) to the wells at the end of the assay, these were incubated at RT until a clear gradation in colour could be seen from the highest to the lowest standard, at which point the reaction was stopped by adding 1 M phosphoric acid (Sigma). All absorbance readings were obtained using a using a FLUOstar Omega plate reader and readings at 540 nm were subtracted from absorbance readings at 450 nm to correct for optical imperfections of the plate. All ELISAs included blanks and standard curves with at least 7 points within range. Sample values were obtained by interpolation into the standard curve; any samples with values out of the range of the standard were assayed again at a dilution(s) that ensured the value was within range.

**Albumin.** All reactions were done in a total volume of 200 μl in an untreated 96-well plate (Greiner Bio-One) to allow vigorous shaking after reagent addition. Samples and standard were diluted 1:5 in PBS

(0.01 M, pH 7.4) and 5 μl were added per well; 5 μl of double-distilled water were added to the blank. Subsequently, 195 μl of stock working solution (prepared according to the manufacturer's instructions) were added into each well, mixed thoroughly using a plate mixer and then left to stand at RT for 10 min before measuring the absorbance of each well at 628 nm using a FLUOstar Omega plate reader.

**Creatinine.** Serum samples did not require deproteinization. The assay was performed using the colorimetric reaction mix, and readings were acquired by measuring absorbance at 570 nm using a FLUOstar Omega plate reader. High levels of haemolysis lead to high background due to the colour; background control samples (i.e., sample with no creatininase) were essential in those cases. 40 μl of serum sample and 10 μl of assay buffer were used per sample well; standards were prepared according to the manufacturer's instructions. After interpolating into the standard curve and correcting for sample dilution, the values are given as nmol/ 50 μl sample; these were transformed to mg/dL, where the molecular weight of creatinine is 113.12 mg/mmol.

### Statistics and reproducibility

Statistical analysis was carried out in GraphPad Prism version 10.4.1 for Windows and version 10.4.0 for Mac (GraphPad Software, Boston, Massachusetts, USA, www.graphpad.com). Power calculations (using G*power) were used to estimate sample size for in vivo animal studies[47]; no statistical method was used to predetermine sample size for functional analysis post sera acquisition. No data were excluded from the analyses. For each in vivo experiment, mice were randomly assigned to experimental groups by the technicians in the facility. Investigators were not blinded to the allocation during experiments and outcome assessment.

Data were analysed for normal distribution based on D'Agostino–Pearson or Shapiro–Wilk normality tests before performing any statistical analyses. Normally distributed data was analysed with the parametric tests detailed in the figure legends; indicated non-parametric tests were applied to data that failed to meet the normality criteria. When more than three comparisons were made, $p$-values were adjusted for multiple comparisons as indicated in the figure legends (Dunnett's post hoc test). For ELISAs, two or three technical replicates were used to estimate experimental mean. Statistical significance was only determined in datasets where each group had three or more biologically independent repeats. A $p$-value $< 0.05$ is considered statistically significant. Each parameter was measured only once in each distinct sample, but more than one parameter could be measured in the same sample (data presented in Fig. 3).

### Data visualisation

Graphs were plotted and visualised in GraphPad Prism version 10.4.1 for Windows and version 10.4.0 for Mac (GraphPad Software, Boston, Massachusetts, USA, www.graphpad.com). Schematics in main figures Figs. 1B and 7 and Supplementary Fig. 6D were made using some images from BioRender (licenses detailed in figure legends). Images were edited and compiled into figures in Affinity Designer V2. (Serif Europe Ltd).

### Reporting summary

Further information on research design is available in the Nature Portfolio Reporting Summary linked to this article.

### Data availability

All plasmid vector sequences are publicly available at [https://doi.org/10.6084/m9.figshare.c.7869659][41]. The previously published genome sequences used in this work are available under accession codes FN543502 and CP009208.1 [https://www.ncbi.nlm.nih.gov/nuccore/CP009208.1]. The data that support this study are available in the Supplementary figures, supplementary tables and Source Data. Source data are provided with this paper.

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

## Acknowledgements

We would like to thank Professor Richard Strugnell and Dr Jonathan Wilksch at the University of Melbourne for providing the reference K2 antisera and C13 monoclonal antibody for use in this work. We would like to thank Professor Matthew Upton and Dr Katie Muddiman at the University of Plymouth for the EC958 isolate. We would like to thank our colleagues Dr Abigail Clements, Dr Andrew Conway Morris, Professor Victor de Lorenzo, Professor Gordon Dougan and Professor Jose Penadés for their critical assessment and feedback in writing this manuscript. We would like to thank our colleagues Dr Mei Chong and Mr Edmond Yau in the Imperial College London Enterprise team for their support in the patent application covering the technology described in this manuscript. This project has been supported by grants from The Royal Society (IC160080) and Medical Research Council (MR/R02671) awarded to GF.

## Author contributions

Conceptualization: J.L.C.W., J.S.-G. and G.F. Methodology: J.L.C.W. and J.S.-G. Formal analysis: J.L.C.W., J.S.-G. Investigation: J.L.C.W., J.S.-G., J.R., J.B. and V.M. Writing–original and draft: J.L.C.W., J.S.-G. and G.F. Writing review and editing: J.L.C.W., J.S.-G. and G.F. Visualisation: J.L.C.W. and J.S.-G. Supervision: J.L.C.W. and G.F. Funding acquisition: G.F.

## Competing interests

The technology underlying citrOgen has a patent pending. Patent applicant: Imperial College Innovations Limited. Application number WO/2025/133638. Name of inventors: Wong, Joshua Liang Chao; Frankel, Gad Meir; Garrido, Julia Sanchez. Status of application: published, patent pending. The *Citrobacter rodentium* chassis, EspO overexpression, lipopolysaccharide, capsular polysaccharide and complex protein antigen presentation and ler-regulon (map) controlled expression are covered in claims 1–57 in WO/2025/133638. The remaining authors declare no competing interests.
