## [Transparent Peer review file · Nature Communications]

citrOgen: a synthesis-free polysaccharide and protein antigen-presentation to antibody-induction platform

Corresponding Author: Dr Gad Frankel

Version 0:

Reviewer comments:

Reviewer #1

(Remarks to the Author)

Given that existing antibody generation technologies rely on the purification or synthesis of antigens, which are time-consuming and costly, this study utilizes the live pathogen *Citrobacter rodentium* (CR) as a natural antigen factory to achieve in situ production and presentation of complex polysaccharide and protein antigens. The study also overexpresses the *pespO* gene to further enhance bacterial colonization and antibody production levels. Three key antigens of the WHO priority pathogen *Klebsiella pneumoniae* (KP) (O1 LPS, K2 capsular polysaccharide, T3F fimbriae) were selected as prototypes, covering major types of bacterial surface antigens, which powerfully demonstrates the universality of the platform. This method has clear advantages, as CR continuously expresses foreign antigens during intestinal infection in mice, closely mimicking the immune stimulation patterns of real pathogens, which may induce a more comprehensive antibody repertoire (e.g., antibodies targeting conformational epitopes). It bypasses the technical challenges of polysaccharide/membrane protein purification (such as the toxic separation of LPS and maintaining the natural conformation of fimbrial proteins).

However, there have been many studies using engineered bacteria for the generation and presentation of polysaccharide and protein antigens, and the authors' argument for the superiority of the described citrOgen strategy is not yet fully substantiated. Specifically, the following points need to be addressed:

1. How do the levels or abundance of the produced antibodies compare with traditional polysaccharide antigen extraction methods?
2. Various other bacteria can also serve as chassis cells to synthesize foreign multi-antigens, such as attenuated *Salmonella* and *Shigella*. How does CR's capability to promote antibody production against polysaccharide antigens compare to these strains? Are there experimental comparisons or data available?
3. What is CR's expression capability for O antigens from other Gram-negative bacteria (e.g., *E. coli* O157, *Vibrio cholerae* O139)?
4. For the application of this study in detection and treatment, monoclonal antibodies must be used. All experiments in this manuscript were conducted using polyclonal antibodies from serum for analysis and identification. The authors should further isolate monoclonal antibody cell lines and confirm that their specificity and affinity reach high levels.

There are some minor comments:

1. There is an extra period after "CRWT controls." in line 410.
2. The notation of statistical differences in multiple places throughout the manuscript is inconsistent. For example, "exact $p=0.0002$ " in line 411 and " $p^{**}=0.0002$ " in line 424 have differing formats. It is recommended to adopt a uniform format and consistently define the meanings of **** and ***. If there is a need to indicate exact p values, they can be marked directly on the figures.

Reviewer #2

(Remarks to the Author)

While the use of heterologous expression systems for bacterial virulence determinants is not a new concept, the idea of using such a system to ensure folding and authentic presentation to enable therapeutic antibody development looks original.

The choice of *Citrobacter* spp. as the expression vehicle, at first glance, would appear to be unlikely. *Citrobacter rodentium* (CR) is a close relative of *Escherichia* with the very rich toolset enabling high level production of proteins, the fact that it is virulent for mice from which monoclonal antibodies can be generated and where it causes a protracted acute infection removes some of the need for accurate protein re-folding.

Antibodies raised through natural infection with the recombinant *Citrobacter* spp. and the infected animals can be tested for infection impact of the antibodies raised against the heterologous targets. This testing of determinants for vaccine potential without antigen expression, purification, refolding and adjuvant formulation adds considerable appeal to the system.

The authors have validated the system using both carbohydrate and protein antigens, principally from the ESKAPE pathogen *Klebsiella pneumoniae* (Kp) and tested the ability of heterologously expressed *Klebsiella* antigens to induce immune responses that were active against *K. pneumoniae*. These antigens include capsular antigens, LPS antigens and the Type III Fimbriae of Kp known as the Mrk fimbriae.

The bacterial engineering is first class – it might be a little surprising that the CR rfa pathway-generated Lipid-A and core of CR can act as a scaffold for the presentation of complete Kp O antigen and the generalisability of this phenomenon, using an O antigen from *Salmonellae* or *Shigellae*, might demonstrate wider application (suggestion 1).

Q. 1 How much is known about the LPS core of CR and how does this core relate to that of other Enterobacteriaceae? The cartoon in Figure 1B could be supported by the actual structure of the core (suggestion 2)

In the *in vivo* studies, it was noted that the CR expression Kp O antigen colonised at lower levels.

Q. 2 Is there any (known) regulation of the O antigen in CR to enable stable pedestal formation? The further engineering (generating more EspO) of the CR to overcome deficiencies induced by heterologous expression of the Kp O antigen by CR was ingenious and demonstrates a very deep understanding of the carrier bacterium (CR).

Q. 3 Is the overexpression of EspO known to increase the immunogenicity of wild type (WT) CR? The infection work to address the role of heterologous O antigen expression in model Kp infections is done well, and the development of suitable biomarkers is an important step in moving away from death of the animals as an endpoint.

Q. 4 In section starting Line 203, it wasn't clear whether the *pespO* plasmid and expression was needed – like with LPS, were preliminary experiments conducted which showed that adhesion to tissues was reduced by capsule production?

Q. 5a Does WT CR express a polysaccharide capsule? One would assume that there would be steric hindrance and reduced access of the T3SS 'needle' to the host cell membrane caused by capsule expression? K2 like many capsules is regulated by genes outside of the locus. Q5b. What were the relative levels of capsule expression between the CRKPK2 and Kp WT? Was the amount of capsule produced by the Kp and the CR roughly the same?

The validation of the anti-K2 capsule antibodies through use of a panel of well characterised strains was done well.

The work on the Mrk fimbriae was also conducted in a thoughtful way and the use of a native CR promoter obviated the requirement for MrkH, the transcriptional activator of native MrkA, and the strict regulation of the levels of cyclic-di-GMP during the *in vivo* analyses.

In the discussion, the fitness cost of heterologous O antigen expression is well documented in the paper, but the fitness cost of making Mrk fimbriae is putative at best.

Q6. Were experiments done showing that native MrkH-regulated Mrk production didn't work well in CR? Kp are known to have a wide array (i.e. >10) of cyclases and PDEs that regulate cyclic-di-GMP levels – is this true for CR as well?

Q. 7 It would have been interesting to see if heterologous antigen expression changed the kinetics of CR fecal shedding, i.e. beyond Day 8? Were these shedding experiments done as a time curve (for example)?

The figures are generally clear – Figure 1E could be a little brighter for the KP O1 Ag picture and the use of double staining e.g. a brighter vital dye (what is FM 4-64?) might have been slightly more convincing.

Reviewer #5

(Remarks to the Author)

In this work, Wong and Sanchez-Garrido, et al., describe the use of the enteric mouse pathogen *Citrobacter rodentium* (CR) as a vehicle for the expression and production of heterologous antigens and functional antibodies, respectively. Specifically, they demonstrate that expression of distinct *Klebsiella pneumoniae* (KP) antigens —O-antigen (O1), capsular polysaccharide (K2), and type 3 fimbriae (T3F)— in *C. rodentium* leads to production of antibodies against the heterologous antigens during host infection. Further, they show evidence that these antibody responses are functional for protection against KP challenge, capsular serotyping, and biofilm inhibition. The work was well designed and conducted, results and figures are clear, and the text is well-written. The system has also potential for expressing other heterologous antigens to produce target-specific antibodies.

Main comment:

My main point has to do with the novelty of the concept itself as expression of heterologous antigens (including O-antigen) and induction of protective, target-specific antibodies have been broadly studied and reported for other enteric pathogens. For example, in *Salmonella Typhimurium* only, heterologous expression of O-antigen from *Salmonella Choleraesuis* or *Pseudomonas aeruginosa* induced target-specific IgG/IgA antibodies during infection, which were protective against enteric and pulmonary challenge by these pathogens, respectively (PMC5533773, PMC529127, PMID: 27476047). Likewise, heterologous expression of an antigen from *Bordetella pertussis* in *Salmonella* conferred protection against tetanus (PMID: 1368983). Thus, authors should discuss the findings and advantages, but also limitations, of the CR model in the context of other bacterial heterologous expression systems used for similar purposes.

Results (General comments):

Do the O1, K1 and T3F antigens have the same structure, composition and/or cell location/abundance when expressed in CR versus naturally in KP? This is relevant for potential future applications of the system, and particularly important for KP O-Ag that is differentially transported and potentially modified in CR.

What subtypes of IgG antibodies are generated against KP O1, K2 and T3F when expressed in CR during infection? Are these the same or different subtypes than those produced against the factors during KP infection?

Because heterologous expression of KP O1 reduced CR colonization, authors overexpressed EspO to increase pathogen attachment and potentially, antigen presentation. Does heterologous expression of KP K2 and T3F also reduce CR colonization? Authors should include these results to confirm whether this is a general or O1-specific effect, and to determine whether overexpression of EspO is necessary for the system.

Figures (Specific points):

Fig. 1C. An LPS/O-Ag extraction gel should be included in addition to the Western blot. Also, the anti-KP O1 antibody seems to detect different O-Ag bands in CR expressing KP O1 than in KP WT. Does this antibody recognize O-Ag specifically but not Lipid A/Core? If so, does this mean the KP O1 structure is different when expressed in CR? This is important to determine as a hybrid O-Ag might lead to less specific antibodies to the target antigen.

Fig. 3. Data show that enteric infection with CR expressing KP O1 conferred protection against KP challenge, potentially due to production of protective, target-specific antibodies. If this is the case, do direct administration of CR KP-O1 immune serum and/or purified pAbs also confer protection? Likewise, do CR KP-K1 or CR KP-T3F infection, immune serum and/or purified pAbs also protect against KP? Finally, lung histology images and pathological scores would be helpful to validate or complement the measurement of the disease markers shown.

Fig. 4E. KP KL2 strains show distinct binding levels for the heterologous (CR) KP K2 antisera. Is this differential pattern also observed with the reference KP K2 antisera? For specificity comparisons, authors should test the seven KP KL2 strains and controls (KP WT, delta-wcaJ) with the reference KP K2 antisera as well.

Fig. 5G & H. A control group treated with the Anti-KP MrkA reference antibody is also required for comparison of efficiency of biofilm inhibition with the CR KP-T3F antisera.

Methods:

L.723, L.741, L.760. C57Bl6/J mice are mentioned multiple times in the methods ('animal work', 'CR infection by oral gavage' & 'cIEC isolation and flow cytometry' subsections). However, only CD-1 mice are described for all mouse experiments in the results. Were both types of mice used? If so, in which specific experiments? What was the rationale of using one type or the other? Were similar results obtained in both CD-1 and C57Bl6/J mice?

L.553. As a common practice, all primer sequences used in the study should be included.

Minor points:

L.250. A rationale for using the map promoter over other LEE promoters for T3SS/Antigen co-regulation would be helpful. Likewise, this part (and the discussion section) highlights that co-expression of antigens and T3SS is required, or at least highly beneficial, for effective antigen presentation and antibody production. However, this T3SS/Antigen co-regulation was done for T3F only, while O1 and K2 induced functional antibodies when expressed in CR from their natural promoters. Thus, authors should describe the co-regulation as an alternative, not a requirement, which can be useful for antigens with complex regulation systems, for example, those requiring additional regulatory genes/operons absent in CR (as T3F) and/or those with low or short-time expression.

L.252 & L.535. A brief description of the presence of Tn7 insertion sites (attTn7) downstream of the glmS gene would be helpful for the non-expert reader to understand why this site was used for insertion of heterologous antigens.

Version 1:

Reviewer comments:

Reviewer #1

(Remarks to the Author)

The authors have addressed my comments.

Reviewer #2

(Remarks to the Author)

The author has answered all my questions, with one possible exception. It relates to this comment.

CRWT encodes a colanic acid biosynthesis locus encoding the genes for the synthesis of this extracellular polysaccharide. The role of this extracellular polysaccharide/capsule is unknown in CR. However, the T3SS needle is required for pathogenesis, or CR fails to colonise the host (PMID: 33707240, Fig 1E). It is important to note that the T3SS is unique, as the needle is extended by a long EspA filament, which would project beyond the capsule. This information has been added at line 239-240. Indeed, in vivo CRKPK2+pespO colonises to the same extent as CRWT so functionally the T3SS must be able to access the host membrane.

While the variant of EspA present in CR may indeed generate a T3SS 'needle' that is sufficiently long as to protrude through the capsule, without measuring both it is hard to be definitive about this. The biology of CR is such that the T3SS is essential and, if the expression of the capsule resulting in an inability of the T3SS to engage with the cell membrane by virtue of steric hindrance from the capsule, then this might affect the ability of the system to deliver the antigens.

Having said this, the fact that it worked suggests that the CR EspA T3SS needle "LIKELY projected further than any capsule comprised of the K2 polysaccharide".

Reviewer #5

(Remarks to the Author)

The authors have successfully addressed all my points. No additional comments/suggestions from my end.

made.

Rebuttal Wong & Sanchez-Garrido et al. 'citrOgen: a synthesis-free polysaccharide and protein antigen-presentation to antibody-induction platform'

Reviewer 1

Reviewer #1 (Remarks to the Author):

Given that existing antibody generation technologies rely on the purification or synthesis of antigens, which are time-consuming and costly, this study utilizes the live pathogen *Citrobacter rodentium* (CR) as a natural antigen factory to achieve in situ production and presentation of complex polysaccharide and protein antigens. The study also overexpresses the *pespO* gene to further enhance bacterial colonization and antibody production levels. Three key antigens of the WHO priority pathogen *Klebsiella pneumoniae* (KP) (O1 LPS, K2 capsular polysaccharide, T3F fimbriae) were selected as prototypes, covering major types of bacterial surface antigens, which powerfully demonstrates the universality of the platform. This method has clear advantages, as CR continuously expresses foreign antigens during intestinal infection in mice, closely mimicking the immune stimulation patterns of real pathogens, which may induce a more comprehensive antibody repertoire (e.g., antibodies targeting conformational epitopes). It bypasses the technical challenges of polysaccharide/membrane protein purification (such as the toxic separation of LPS and maintaining the natural conformation of fimbrial proteins).

However, there have been many studies using engineered bacteria for the generation and presentation of polysaccharide and protein antigens, and the authors' argument for the superiority of the described citrOgen strategy is not yet fully substantiated. Specifically, the following points need to be addressed

We thank the reviewer for their comments and the recognition of the powerful 'universality' of the platform. This represents a key advance which differentiates our work from others, together with the rapid nature in which we can engineer citrOgen strains and obtain specific antibody responses.

1. How do the levels or abundance of the produced antibodies compare with traditional polysaccharide antigen extraction methods?

O-antigens are putative vaccine candidates in KP. One multivalent glycoconjugate candidate 'Kleb4V' (encompassing KP O1, O2, O2afg and O3b) was generated by LimmaTech (LimmaTech Biologics AG, previously Glycovaxyn AG). However, we are unable to obtain Kleb4V (now fully owned by GSK as a proprietary product) to test head-to-head with citrOgen CR_{KPO1} in mice.

Additionally, other glycoconjugates have been tested in mice. This includes an O1-EPA glycoconjugate vaccine (PMID: 37146068) which was administered with two booster doses (doses at day 0, 14 and 28). This vaccine failed to induce anti-KP Abs able to bind a highly related KP strain (ATCC43816, the parental strain to that used in our study), even at day 42 post initial vaccination. However, when the authors generated an acapsular isogenic mutant of ATCC43816, they did observe anti-KP antibody binding (mean c. 75ng/ml at 42 days). We obtain a 10-fold higher anti-KP antibody titre (mean c. 750ng/ml, Figure 2C) able to bind the capsulated (natural) KP strain in 30 days.

K-antigens are also putative vaccine candidate in KP. In PMID 37146068 the authors also generated a K2-EPA vaccine, also administered with two booster doses to mice. At 42 days post initial vaccination the anti-KP antibody binding response was c. 75ng/ml. Our titre following a single CR_{KPK2} administration at 30dpi was c. 1000ng/ml.

These glycoconjugates are unavailable for us to test directly in head-to-head comparisons.

MrkA is a putative vaccine candidate in KP. The core of the invention we present is the ability to produce the entire type 3 fimbrial structure (and not just MrkA). In PMID: 26768253, a MrkA vaccine (combination of monomeric and oligomeric MrkA with Freund adjuvant) was generated and tested by MedImmune. This (15ug protein) was administered with two booster doses (doses at day 0, 14 and 28) to mice. Antibody titres were not reported in PMID: 26768253 so we are unable to compare responses to CR_{T3F}.

In summary, in all cases, citrOgen strain infection results in more rapid specific heterologous antigen antibody responses and, uniquely, encompasses both polysaccharide and protein antigens.

2. Various other bacteria can also serve as chassis cells to synthesize foreign multi-antigens, such as attenuated Salmonella and Shigella. How does CR's capability to promote antibody production against polysaccharide antigens compare to these strains? Are there experimental comparisons or data available?

We thank the reviewer for this question. We answer this in the section dedicated to Reviewer 5 (as there are direct citations provided that we address and to prevent repetition in the rebuttal).

3. What is CR's expression capability for O antigens from other Gram-negative bacteria (e.g., *E. coli* O157, *Vibrio cholerae* O139)?

This comment prompted us to test the citrOgen platform using another O-antigen. We chose O25b, a common O-antigen encoded by ST131 *E. coli*, which cause urinary tract infection and bacteraemia. Moreover, it also enabled us to compare citrOgen to traditional methods used by others (to complement response to comment 1). Like KP, *E. coli* ST131 is a multidrug-resistant target for contemporary mAb therapy and indeed, vaccine design (PMID: 35311580). The work in this publication was conducted by Pfizer and the O25b-CRM glycoconjugates are not available for us to test head-to-head with our newly generated strain CR_{ECO25b}. Whilst generation of antibodies using the O25b-CRM glycoconjugate required the generation of 4 different glycoconjugate scaffolds, two booster doses and a >91-day immunisation course, citrOgen enables the generation of specific anti-O25b antibody responses in 35 days. The added value to this experiment lies in that O25b export is conducted via the wzy-dependent pathway instead of the ABC transporter-dependent polysaccharide export pathway used by KP.

We include new data in Figure 4 and Lines 208-227, which expands the citrOgen O-antigen section before moving onto capsular antigens. We were able to generate these specific anti-O25b antibody responses in-line with the workflow and timelines outlined in Figure 7. This further supports our assertion that the presented technology is adaptable and rapid.

4. For the application of this study in detection and treatment, monoclonal antibodies must be used. All experiments in this manuscript were conducted using polyclonal antibodies from serum for analysis and identification. The authors should further isolate monoclonal antibody cell lines and confirm that their specificity and affinity reach high levels.

We agree with the reviewer that this represents the next step in further evaluating our technology. However, generating and testing mAbs is beyond the scope of this manuscript. In this study we addressed the bottleneck of antigen preparation. The majority of mAb research has focussed on memory B-cell isolation techniques, Fc-effector function modification/engineering and therapeutic testing of efficacy. To that end, entire manuscripts have been dedicated to isolating, generating and testing mAbs against KP O1 and O2 in this journal (e.g. PMID 29222409 & PMID: 39285158). Instead, our goal was to provide extensive data supporting the rapid generation of specific antibody responses to multiple antigen types, all presented in their native state, which is at the core of the invention. We think having this contained in a single manuscript is more important than the already optimised downstream process of generating mAbs against the antigens tested. We have now expanded the antigen repertoire to O25b, addressing comment 3 and expanding on our answer to comment 1.

There are some minor comments:

1. There is an extra period after "CRWT controls." in line 410.

Thank you- corrected.

2. The notation of statistical differences in multiple places throughout the manuscript is inconsistent. For example, "exact $p=0.0002$ " in line 411 and " $p^{**}=0.0002$ " in line 424 have differing formats. It is recommended to adopt a uniform format and consistently define the meanings of **** and ***. If there is a need to indicate exact p values, they can be marked directly on the figures.

Thank you- adjusted throughout.

Reviewer 2

While the use of heterologous expression systems for bacterial virulence determinants is not a new concept, the idea of using such a system to ensure folding and authentic presentation to enable therapeutic antibody development looks original. The choice of *Citrobacter* spp. as the expression vehicle, at first glance, would appear to be unlikely. *Citrobacter rodentium* (CR) is a close relative of *Escherichia* with the very rich toolset enabling high level production of proteins, the fact that it is virulent for mice from which monoclonal antibodies can be generated and where it causes a protracted acute infection removes some of the need for accurate protein re-folding.

Antibodies raised through natural infection with the recombinant *Citrobacter* spp. and the infected animals can be tested for infection impact of the antibodies raised against the heterologous targets. This testing of determinants for vaccine potential without antigen expression, purification, refolding and adjuvant formulation adds considerable appeal to the system.

The authors have validated the system using both carbohydrate and protein antigens, principally from the ESKAPE pathogen *Klebsiella pneumoniae* (Kp) and tested the ability of heterologously expressed *Klebsiella* antigens to induce immune responses that were active against *K. pneumoniae*. These antigens include capsular antigens, LPS antigens and the Type III Fimbriae of Kp known as the Mrk fimbriae.

The bacterial engineering is first class – it might be a little surprising that the CR *rfa* pathway-generated Lipid-A and core of CR can act as a scaffold for the presentation of complete Kp O antigen and the generalisability of this phenomenon, using an O antigen from *Salmonellae* or *Shigellae*, might demonstrate wider application (suggestion 1).

We thank the reviewer for the complementary summary of our work, which highlights the novelty in our choice of *Citrobacter rodentium* for this platform technology. Indeed, whilst other heterologous expression systems have been proposed, none have been able to cover the breadth of antigens we achieve with CR. To that end, we wanted to ensure that the bacterial engineering techniques we employed were not only contemporary but, more importantly, adaptable. A worked example of the latter has now been added to the manuscript (Lines 208-227 and Figure 4) at the suggestion of both Reviewer 1 and 2. This expands our pathogenic targets to another critical priority organism, ST131 *E. coli* and is described in the responses to Reviewer 1 above in depth.

We thank Reviewer 2 for highlighting that *E. coli* already has a very rich toolset which we have added into the manuscript in the Discussion (Line 355-357) and improved the manuscript.

Q. 1 How much is known about the LPS core of CR and how does this core relate to that of other Enterobacteriaceae? The cartoon in Figure 1B could be supported by the actual structure of the core (suggestion 2).

CR expresses an R2 core (MW 2691 Da) based on KEGG pathway analysis and that *waaO* and *waaR* encoded in the CR *rfa* locus. The molecular weight for this core can be found here <https://pubchem.ncbi.nlm.nih.gov/compound/E.-coli-R2-core>.

The KP used in this manuscript expresses a KP type 1 core (3057 Da) based on the orientation of the genes flanking *waaL* and analysis conducted in PMID: 34431721. KP has two core types (type 1 and 2) and the structures can be found in PMID: 16735743.

Interestingly, the significance of divergent KP core type expression on virulence remains unexplored. We obtained the molecular weight of KP type 1 core in PMID: 11986326.

Given that we have now added and demonstrated that an alternative O-antigen (O25b), with a different O-antigen reducing sugar for ligation (Gal in KP O1, Glu in EC O25b), can ligate to CR R2 core, we feel that this is no longer required and complicates the cartoon. The O25b data suggests that CR R2 core is a good scaffold for further expansion to even more O-antigens, especially as the wzy-export pathway utilised by O25b is more common.

In the in vivo studies, it was noted that the CR expression Kp O antigen colonised at lower levels.

Q. 2 Is there are (known) regulation of the O antigen in CR to enable stable pedestal formation?

The further engineering (generating more EspO) of the CR to overcome deficiencies induced by heterologous expression of the Kp O antigen by CR was ingenious and demonstrates a very deep understanding of the carrier bacterium (CR).

We thank the reviewer for acknowledging the EspO ingenuity. The analytical tools that we have built allowed us to explore the underpinning mechanism leading to restoration of colonisation. CR employs a wzz-mediated chain length determinant mechanism, and this influences O-antigen length. However, the upstream factors governing wzz expression itself are unknown and the impact on O-antigen length on stable pedestal formation in vivo unexplored. However, our data suggests that pedestal formation in vitro is unaffected by O-antigen length (eg Supplementary Fig 1B pedestals KPO1). The O-antigen chain length of EC O25b and KPO1 (in the context of EspO overexpression) results in efficient mouse colonisation, so stable pedestal formation can occur over the chain length differences imposed by O152 (CR_{WT}), KPO1 (CR_{KPO1}) and ECO25b (CR_{ECO25b}).

Q. 3 Is the overexpression of EspO known to increase the immunogenicity of wild type (WT) CR?

No, this is not known, and much type 3 secretion system effector biology has relied on effector deletions to demonstrate phenotypes rather than overexpression. We have examined the convalescent sera from mice orally infected with CR_{WT} vs CR_{WT+pespO} and found no differences in total anti-CR IgG post infection (Rebuttal Figure 1) suggesting that the immunogenicity, as measured by IgG levels, is not enhanced. As we demonstrate in our manuscript that EspO overexpression increases CR attachment to the intestinal epithelial cells, providing a robust explanation of mechanism, we have not included the IgG data in the manuscript.

Figure 1: CR_{WT} overexpressing EspO does not induce higher anti-CR IgG levels in serum at 30 dpi compared to those generated as a result of CR_{WT} infection. Results obtained from an ELISA using heat-killed CR_{WT}. Mean \pm SEM from n=8-10 mice, Welch's unpaired t test. Mock infected (PBS) provided as baseline control.

The infection work to address the role of heterologous O antigen expression in model Kp infections is done well, and the development of suitable biomarkers is an important step in moving away from death of the animals as an endpoint.

We thank the reviewer for this point, which is of great importance whilst evaluating severe infection in translational models. Indeed, as recognised by the reviewer, this approach moves away from using death as an end-point- this is ethically important but also aids in the reproducibility of data. For example, we are based in the United Kingdom where death cannot legally be used as an endpoint precluding us from confirming results published from groups working in countries where the law permits this work.

In addition, these outcome variables are used by clinicians to test efficacy of therapeutics in human patients, improving translatability of preclinical studies.

Q. 4 In section starting Line 203, it wasn't clear whether the pespO plasmid and expression was needed – like with LPS, were preliminary experiments conducted which showed that adhesion to tissues was reduced by capsule production?

Thank you for this comment which also raised by Reviewer 5. We have now conducted these studies which demonstrate that K2 expression does negatively affect colonisation. This is covered in the text (Lines 239-243) and Supplementary Figures 5B.

Q. 5a Does WT CR express a polysaccharide capsule? One would assume that there would be steric hindrance and reduced access of the T3SS 'needle' to the host cell membrane cause by capsule expression? K2 like many capsules is regulated by genes outside of the locus.

CR_{WT} encodes a colanic acid biosynthesis locus encoding the genes for the synthesis of this extracellular polysaccharide. The role of this extracellular polysaccharide/capsule is unknown in CR. However, the T3SS needle is required for pathogenesis, or CR fails to colonise the host (PMID: 33707240, Fig 1E). It is important to note that the T3SS is unique, as the needle is extended by a long EspA filament, which would project beyond the capsule. This information has been added at line 239-240. Indeed, *in vivo* CR_{KPK2+pespO} colonises to the same extent as CR_{WT} so functionally the T3SS must be able to access the host membrane.

Q5b. What were the relative levels of capsule expression between the CRKPK2 and Kp WT? Was the amount of capsule produced by the Kp and the CR roughly the same?

Our goal was to raise anti-KPK2 antibodies via heterologous expression in citrOgen *in vivo* and this was achieved. Furthermore, we do qualitatively assess expression by immunostaining in Supplementary Figure 5A and strong K2 staining is observed in CR_{KPK2}. If our approach had failed i.e. CR_{KPK2} did not induce anti-KPK2 antibodies then we agree with the reviewer's comment, and we would have proceeded to quantify K2 concentrations. The assays for K2 are based on uronic acid quantification *in vitro*, not *in vivo*, and are only appropriate for looking at large changes which also limits their utility in accurate determination of the capsular antigen.

We have expanded the legend of Figure 7 on QC that can be achieved in the workflow. We thank the reviewer for this point which has allowed us to improve the legend (Lines 575-5177).

The validation of the anti-K2 capsule antibodies through use of a panel of well characterised strains was done well.

The work on the Mrk fimbriae was also conducted in a thoughtful way and the use of a native CR promoter obviated the requirement for MrkH, the transcriptional activator of native MrkA, and the strict regulation of the levels of cyclic-di-GMP during the *in vivo* analyses.

We thank the reviewer for this comment.

In the discussion, the fitness cost of heterologous O antigen expression is well documented in the paper, but the fitness cost of making Mrk fimbriae is putative at best.

We have now established with additional experiments that Mrk fimbriae expression does result in a fitness cost in CR. This has been added to the text in lines 296-298 and 6ementary Figure 5C.

Q6. Were experiments done showing that native MrkH-regulated Mrk production didn't work will in CR? Kp are known to have a wide array (i.e. >10) of cyclases and PDEs that regulate cyclic-di-GMP levels – is this true for CR as well?

These experiments were not conducted. We decided *a priori* to generate and test an expression system that was tailored to CR physiology and that could be modified for any user's defined antigen. We chose T3F as a prototype in KP as there is a literature base for its antigenicity (PMID: 26768253). We could for example, have chosen KP type 1 fimbriae as an alternative multiprotein surface complex. KP T1F expression is regulated by an invertible phase variable switch. If we chose to clone this into our Pmap driven vector, the exact same approach would have been followed as outlined in the manuscript- the T1F structural gene operon would be cloned alone, removing control of the phase on and off controllers (FimB/FimE).

Q. 7 It would have been interesting to see if heterologous antigen expression changed the kinetics of CR fecal shedding, i.e. beyond Day 8? Were these shedding experiments done as a time curve (for example)?

Initial experiments did address whether the kinetics of CR shedding was affected by heterologous antigen expression, but no significant changes were observed (Rebuttal Figure 2A). We therefore collected faecal samples at the peak of infection (8 dpi) and

to confirm clearance. EspO is demonstrated again to restore colonisation in the context of heterologous antigen (Rebuttal Fig 2B).

The figures are generally clear – Figure 1E could be a little brighter for the KP 01 Ag picture and the use of double staining e.g. a brighter vital dye (what is FM 4-64?) might be have been slightly more convincing.

Thank you for this comment. We have adjusted both the CR_{WT} and CR_{KPO1} image using the same algorithm. With regards to Fig 1E we chose DAPI as our structural marker instead as it provides better resolution of colonic structures without interfering with the bacterial stains. FM4-64 is a styryl dye that will fluoresce in all cell membranes including that of bacteria, used in Figure 5F for this purpose.

Figure 2. A. Paecal shedding of the CR_{WT}, a CR strain expressing a heterologous O-antigen (CR_{OAg}) and CR_{OAg} overexpressing EspO. Graph shows mean from n=10 mice from 2 biological repeats. B. 8dpi EspO overexpression restores colonisation in the context of heterologous antigen expression. One-way ANOVA.

Reviewer 5

In this work, Wong and Sanchez-Garrido, et al., describe the use of the enteric mouse pathogen *Citrobacter rodentium* (CR) as a vehicle for the expression and production of heterologous antigens and functional antibodies, respectively. Specifically, they demonstrate that expression of distinct *Klebsiella pneumoniae* (KP) antigens —O-antigen (O1), capsular polysaccharide (K2), and type 3 fimbriae (T3F)— in *C. rodentium* leads to production of antibodies against the heterologous antigens during host infection. Further, they show evidence that these antibody responses are functional for protection against KP challenge, capsular serotyping, and biofilm inhibition. The work was well designed and conducted, results and figures are clear, and the text is well-written. The system has also potential for expressing other heterologous antigens to produce target-specific antibodies.

Main comment:

My main point has to do with the novelty of the concept itself as expression of heterologous antigens (including O-antigen) and induction of protective, target-specific antibodies have been broadly studied and reported for other enteric pathogens. For example, in *Salmonella Typhimurium* only, heterologous expression of O-antigen from *Salmonella Choleraesuis* or *Pseudomonas aeruginosa* induced target-specific IgG/IgA antibodies during infection, which were protective against enteric and pulmonary challenge by these pathogens, respectively (PMC5533773, PMC529127, PMID: 27476047). Likewise, heterologous expression of an antigen from *Bordetella pertussis* in *Salmonella* conferred protection against tetanus (PMID: 1368983). Thus, authors should discuss the findings and advantages, but also limitations, of the CR model in the context of other bacterial heterologous expression systems used for similar purposes.

We thank the reviewer for the kind comments regarding study design and execution. We would like to use this opportunity to expand on the Reviewer 1's main comment and add clarification of what we consider the major advances and impact of our work. Most importantly we want to demonstrate how this diverges from the published literature and represents novelty and not incremental advance.

The citations listed are all centred on the use of live attenuated vaccine strains using various attenuation strategies in *Salmonella* (*cya* deletion PMC5533773, *wecA* deletion PMID: 27476047 and *aroA* deletion PMC529127) with the aim of protecting against challenge. Two manuscripts use heterologous expression of LPS O-antigens (*Pseudomonas aeruginosa* O11 and that from *Salmonella choleraesuis*) and one the proteinaceous atoxic fragment of tetanus toxin. Our project's aim is quite different—our goal was to develop a modular, adaptable platform capable of generating specific antibody responses to all major antigenic types following infection with a virulent strain (in mice) and achieve this in a timely manner. Of note, our platform represents the first 'chassis' that has been used to heterologously present a capsular polysaccharide from a different species and a large multiprotein complex that are functional (rather than single protein subunits).

In PMC529127 a great advance is made in that the stability of the plasmid expressing the tetanus toxin is alleviated by changing the promoter to the anaerobic driven promoter *nirB*. Indeed, this change results in higher antibody levels in mice which then protect animals from tetanus toxin administration. Our approach was to use inherently more complex stable chromosomal mutants and drive expression in our bespoke designed and modifiable Pmap system for protein antigen expression. Whilst Chatfield *et al.* were able to induce antibodies to a small single protein subunit, we

express a whole organelle (the type 3 fimbriae). In doing so, downstream isolation of mAbs against different components (e.g. major pillin MrkA and adhesin tip MrkD) can now be achieved in a single citrOgen infection.

In PMC5533773, the authors note that their vaccine strain results in poor heterologous protection due to poor/low *Salmonella choleraesuis* O-antigen expression and/or poor expression vector stability. They suggest that an expression vector with a higher copy number may have improved this. Not only do we generate stable chromosomal mutants but also provide the scientific community with validated vectors and chromosomal insertion sites to support further work (Figure 7). Interestingly, when we expressed KPO1 under native CR *rfb* expression we observe a fitness cost and circumvented this by overexpressing EspO. Thus, increasing expression with a higher copy number vector may further attenuate the *Salmonella typhimurium* vector used by Zhao *et al.* and result in even poorer protection.

In PMID: 27476047 the *Salmonella* Typhimurium vector expressing PA O11 failed to significantly protect mice from subsequent challenge with a lethal dose. Responses were different when the vector was administered by intraperitoneal infection, highlighting the complexity and antigen-dependent nature of using *Salmonella* Typhimurium as a vector system. However, in citrOgen we show that, independent of antigen choice, routine administration of CR to the gut results in systemic antibody levels to heterologous antigens. We have also now expanded this finding to show that beyond KP antigens we can raise antibodies to EC O25b. This is a major revision to the manuscript.

Whilst we do use protection as a read-out for functional antibody responses in the case of KPO1 our goal was not to develop a vaccine platform. To that end CR does not infect humans and is host restricted to mice. We appreciate that this confusion has arisen from our failure to address previous heterologous expression work in the introduction. We thank the reviewer for their main comment which we have now used to improve our manuscript. To that end, we now address this important point in an expanded introduction lines 31-45 and 80-89.

We note that Reviewer 2 commented on the novelty in choice of CR. The choice of pathogen appears to be key as approaches with attenuated *Salmonella* and *Shigella* strains showing promise in the vaccine literature have failed human translation. A case in point was the use of *Salmonella* Ty21a to express a *Shigella* O-antigen, which raised no heterologous antigen antibodies when administered to humans in a clinical trial. Our system does not rely on the ability to also infect human hosts like vaccine platforms and takes advantage of the fact that CR clearance in mice is known to be dependent on IgG, unlike what is seen with intracellular pathogens like *Salmonella* Typhimurium.

Results (General comments):

Do the O1, K1 and T3F antigens have the same structure, composition and/or cell location/abundance when expressed in CR versus naturally in KP? This is relevant for potential future applications of the system, and particularly important for KP O-Ag that is differentially transported and potentially modified in CR.

We believe this question may be rooted in our failure to address vaccination as outlined above. For each selected antigen we present it on CR and check for heterologous antigen expression. Following infection and clearance, we compare the antibody responses to reference antibodies which are known to specifically bind the

antigen. In each case the pattern of binding by immunofluorescence phenocopies the reference antibody (KPO1 Figure 2E, KPK2 Figure 5D, T3F Figure 6F). In doing so we show that the binding is specific and therefore the antigen presented by CR is correct. To bolster this, we also generated specific mutations in KP of each antigen and demonstrate that sera raised only binds KP_{WT} (KPO1 Fig 2D, KPK2 Fig 5C, T3F Fig 6E). This shows that the CR-presented antigen is immunologically the same as that expressed naturally in KP, given that the antisera produced in response to said antigen only binds to KP_{WT} (and not the mutant that lacks the antigen). The immunofluorescence phenocopying our reference antibodies further supports this with localisation. While we cannot demonstrate the antigens expressed by CR are exactly the same as those expressed by KP, especially in terms of abundance, our aim was to generate antibodies to the native conformation that could effectively and specifically bind the antigen in our pathogen of interest, which we have achieved.

What subtypes of IgG antibodies are generated against KP O1, K2 and T3F when expressed in CR during infection? Are these the same or different subtypes than those produced against the factors during KP infection?

We thank the Reviewer for this comment. We chose to assay for IgG as these are the human antibody class that are most widely used as prospective mAb therapeutics (e.g. PMID 29222409 and <https://www.biorxiv.org/content/10.1101/2024.02.14.580141v2.full>). Understanding the class and subclass distribution would be very important in the context of vaccine responses. However, in mAb development, routine class and subclass switching is achieved *in vitro* by subcloning (e.g. PMID: 30667360). Furthermore, we chose total IgG as most anti-bacterial mAbs in development do not have a natural subclass. Instead, they are Fc-engineered (improved effector function/half-life extension), thus no longer in their native state and do not belong to a natural subclass.

Recently, the only study (PMID: 38746292) pertaining to antibody classes induced by these KP antigens following blood stream infection in humans was published. This highlighted the complexity in these antigens for vaccine design with different responses noted even with different KP O-antigens (e.g. KP O1 vs KP O3b). Here, class was assessed (IgG, IgM and IgA) but subclasses were not investigated. IgG was universally raised to O-antigen and MrkA. K2 was not assessed as there is a wide range of K-antigens in the KP species. On balance, in light of this new data, assaying IgG appears to be a good measure of antibody induction.

Because heterologous expression of KP O1 reduced CR colonization, authors overexpressed EspO to increase pathogen attachment and potentially, antigen presentation. Does heterologous expression of KP K2 and T3F also reduce CR colonization? Authors should include these results to confirm whether this is a general or O1-specific effect, and to determine whether overexpression of EspO is necessary for the system.

We thank the reviewer for this comment and have addressed with extra supporting experiments as described in the Reviewer 2 Question 4 section.

Figures (Specific points):

Fig. 1C. An LPS/O-Ag extraction gel should be included in addition to the Western blot. Also, the anti-KP O1 antibody seems to detect different O-Ag bands in CR expressing KP O1 than in KP WT. Does this antibody recognize O-Ag specifically but not Lipid A/Core? If so, does

this mean the KP O1 structure is different when expressed in CR? This is important to determine as a hybrid O-Ag might lead to less specific antibodies to the target antigen.

The KPO1 antibody is a mAb that specifically recognises KPO1 O-antigen and not core regions (PMID: 18725260). The O-antigen structure in CR_{KPO1} must therefore contain this epitope. We know that the antibodies raised following CR_{KPO1} infection are specific as there is no binding to the KP_{Δrfb} strain (Figure 2D), a strain that maintains core expression without any KPO1 ligated to it.

We have now also added a silver stained LPS gel in Supplementary Figure 1A.

Fig. 3. Data show that enteric infection with CR expressing KP O1 conferred protection against KP challenge, potentially due to production of protective, target-specific antibodies. If this is the case, do direct administration of CR KP-O1 immune serum and/or purified pAbs also confer protection? Likewise, do CR KP-K1 or CR KP-T3F infection, immune serum and/or purified pAbs also protect against KP? Finally, lung histology images and pathological scores would be helpful to validate or complement the measurement of the disease markers shown.

The aim of this experiment was to demonstrate that the antibodies generated by citrOgen are functional and can recognise the pathogen in vivo. Using the pAb prophylactically or therapeutically would constitute a project on its own. Indeed, we are working on mAb prophylaxis against KP and the optimisation phase lasted 12 months.

We did not conduct histology in the lungs. To enable this, we would need to conduct fixative inflation of whole respiratory system via the trachea. This would preclude measuring lung bacterial burdens (Fig 3A), lung homogenate IL1b (Fig 3K) and lung homogenate MPO (Fig 3M) which require lung homogenate preparation (without fixative present). We prioritised these more clinically relevant outcome variables in assessing protection. KP causes a lobar pneumonia which can develop in any of the multiple mouse lobes between animals infected with the same inoculum and methodology. Whilst we could have selected a single lobe for histopathology and processed the remainder for homogenate measurements, this would fail to satisfactorily measure the global burden of disease within the lung tissue.

Fig. 4E. KP KL2 strains show distinct binding levels for the heterologous (CR) KP K2 antisera. Is this differential pattern also observed with the reference KP K2 antisera? For specificity comparisons, authors should test the seven KP KL2 strains and controls (KP WT, delta-wcaJ) with the reference KP K2 antisera as well.

When using our reference KP K2 reference antisera we also observe a differential pattern between the different KL2 strains in the collection, although the binding with this antibody is stronger overall (higher MFI values). Notably, we can clearly see much lower binding in MRSN 740795 both with our citrOgen anti-KP K2 and the reference antibody, despite this strain expressing O1v1 like all except 572640 and 599975, which express O1v2. We have now added this data to the supplementary (Supplementary Figure 5D).

Fig. 5G & H. A control group treated with the Anti-KP MrkA reference antibody is also required for comparison of efficiency of biofilm inhibition with the CR KP-T3F antisera

We thank the reviewer for this comment which was in line with our initial thinking when designing this experiment. In CR_{T3F} infection the host sees the entire type 3

fimbriae. This includes all components (MrkA, B, C, D&F). Therefore, antibodies can be raised to each component, a key advantage to citrOgen vs methods where MrkA alone is used as the immunogen (PMID: 26768253). Our reference antibody was raised against recombinant MrkA and not the entire T3F structure. Therefore, it would not act as an appropriate control as it would not have activity against, for example, MrkD, which represents the T3F adhesin. As initial adhesion is a key factor in biofilm formation, we excluded using the reference MrkA antibody as a control. Instead, we chose to use both CR_{WT} sera and KP without any sera as more appropriate controls for this assay (Fig 6G). Unfortunately, we were unable to find any reports or commercial sellers of polyclonal sera raised to the entire purified KP T3F and, to our knowledge, this does not exist.

Methods:

L.723, L.741, L760. C57Bl6/J mice are mentioned multiple times in the methods ('animal work', 'CR infection by oral gavage' & 'cIEC isolation and flow cytometry' subsections). However, only CD-1 mice are described for all mouse experiments in the results. Were both types of mice used? If so, in which specific experiments? What was the rationale of using one type or the other? Were similar results obtained in both CD-1 and C57BL6/J mice?

CD-1 mice are used for all experiments evaluating the citrOgen platform itself; C57BL6/J mice were used for the investigation of the role of EspO during infection as part of another project in the laboratory. This included CR attachment to cells, shown in Supplementary Figure 2C (the specific figure where C57BL/6 mice were used has been added to the methods for clarity, lines 867-868). The C57BL6/J results lead us to evaluate if EspO overexpression could ameliorate the fitness cost imposed by heterologous antigen expression in this work. We performed the CR attachment experiment once with CD-1 mice to confirm if the C57BL/6 was reproducible in the CD-1 background; indeed this was the case (Rebuttal Figure 3). Further, our immunofluorescence staining in 2B confirmed our results in the context of a CR_{KPO1}.

Figure 3. The proportion of CR+ IECs isolated from the distal colon is higher in CD-1 mice infected with CR+pespO. Graph shows mean±SEM from 1 biological repeats.

L.553. As a common practice, all primer sequences used in the study should be included.

This are now included in the online repository of genbank files and publicly accessible.

Minor points:

L.250. A rationale for using the map promoter over other LEE promoters for T3SS/Antigen co-regulation would be helpful. Likewise, this part (and the discussion section) highlights that

co-expression of antigens and T3SS is required, or at least highly beneficial, for effective antigen presentation and antibody production. However, this T3SS/Antigen co-regulation was done for T3F only, while O1 and K2 induced functional antibodies when expressed in CR from their natural promoters. Thus, authors should describe the co-regulation as an alternative, not a requirement, which can be useful for antigens with complex regulation systems, for example, those requiring additional regulatory genes/operons absent in CR (as T3F) and/or those with low or short-time expression.

We chose *map* over other promoters because it is one of the first effectors to be translocated in studies with the related pathogen EPEC (PMID: 23900171) and one of the effectors we frequently identify when performing proteomics analysis of intestinal epithelial cells from mice infected with CR (PMID: 33707240). Further, *map* has a defined promoter region and is monocistronic, which avoids complications associated with other T3SS effectors, which lie in operons.

We have now highlighted the advantages of the *map* promoter in the text (Lines 352-354 and legend of Figure 7). This is key for heterologous antigens where expression needs to be regulated, ensuring expression only during infection (i.e. at the same time as T3SS effectors). This strategy also ensures absent expression when not required (e.g. during inoculation preparation) and mitigates against the lack of regulatory genes in CR for future heterologous antigens.

L252 & L.535. A brief description of the presence of Tn7 insertion sites (attTn7) downstream of the *glmS* gene would be helpful for the non-expert reader to understand why this site was used for insertion of heterologous antigens.

Thank you, we have now entered this at the first mention (Lines 120-122)

REVIEWERS' COMMENTS

Reviewer #1 (Remarks to the Author):

The authors have addressed my comments.

Reviewer #2 (Remarks to the Author):

The author has answered all my questions, with one possible exception. It relates to this comment.

CRWT encodes a colanic acid biosynthesis locus encoding the genes for the synthesis of this extracellular polysaccharide. The role of this extracellular polysaccharide/capsule is unknown in CR. However, the T3SS needle is required for pathogenesis, or CR fails to colonise the host (PMID: 33707240, Fig 1E). It is important to note that the T3SS is unique, as the needle is extended by a long EspA filament, which would project beyond the capsule. This information has been added at line 239-240. Indeed, in vivo CRKPK2+pespO colonises to the same extent as CRWT so functionally the T3SS must be able to access the host membrane.

While the variant of EspA present in CR may indeed generate a T3SS 'needle' that is sufficiently long as to protrude through the capsule, without measuring both it is hard to be definitive about this. The biology of CR is such that the T3SS is essential and, if the expression of the capsule resulting in an inability of the T3SS to engage with the cell membrane by virtue of steric hindrance from the capsule, then this might affect the ability of the system to deliver the antigens.

Having said this, the fact that it worked suggests that the CR EspA T3SS needle "LIKELY projected further than any capsule comprised of the K2 polysaccharide".

Thank you for this comment. In line with your thinking we also chose to use the word 'likely' in line 240 of the manuscript.

Reviewer #5 (Remarks to the Author):

The authors have successfully addressed all my points. No additional comments/suggestions from my end.